# MV-Adapter: Multi-view Consistent Image Generation Made Easy

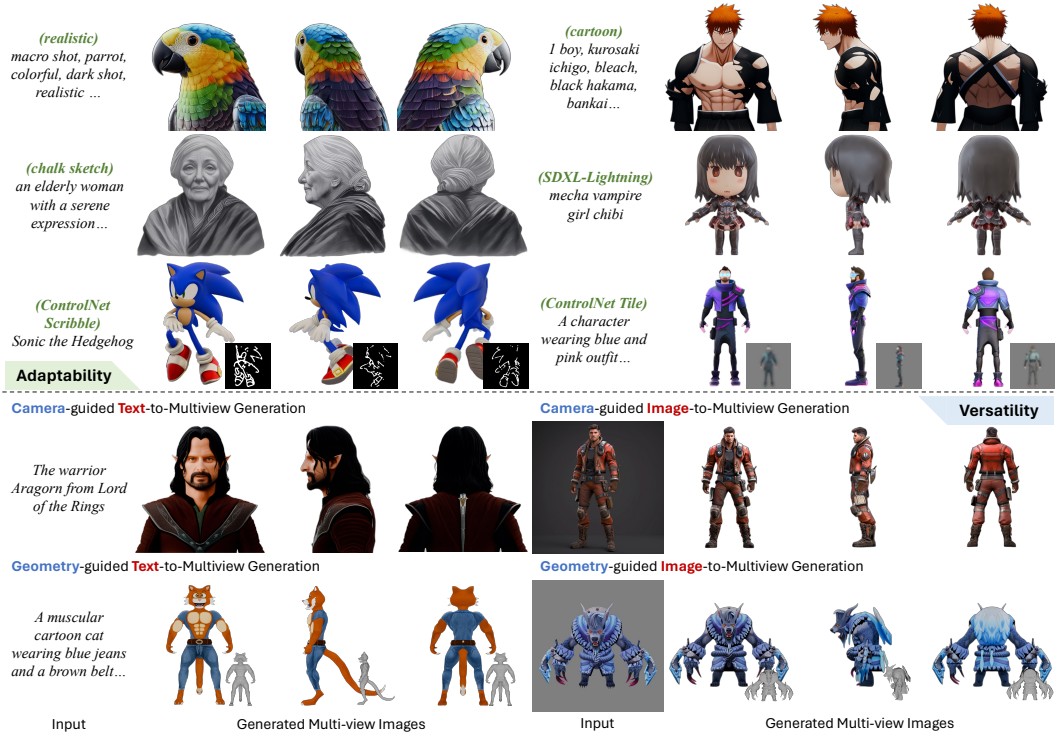

Figure 1: MV-Adapter is a versatile plug-and-play adapter that turns existing pre-trained text-to-image (T2I) diffusion models to multi-view image generators. ***Row 1,2,3***: results by integrating MV-Adapter with personalized T2I models, distilled few-step T2I models, and ControlNets (Zhang et al., 2023), demonstrating its **adaptability**. ***Row 4,5***: results under various control signals, including view-guided or geometry-guided generation with text or image inputs, showcasing its **versatility**.

## ABSTRACT

Generating multi-view images of an object has important applications in content creation and perception. Existing methods achieved this by making invasive changes to pre-trained text-to-image (T2I) models and performing full-parameter training, leading to three main limitations: (1) High computational costs, especially for high-resolution outputs; (2) Incompatibility with derivatives and extensions of the base model, such as personalized models, distilled few-step models, and plugins like ControlNets; (3) Limited versatility, as they primarily serve a single purpose and cannot handle diverse conditioning signals such as text, images, and geometry. In this paper, we present MV-Adapter, a plug-and-play module working on top of pre-trained T2I models. MV-Adapter enables efficient training for high-resolution synthesis while maintaining full compatibility with all kinds of derivatives of the base T2I model. MV-Adapter provides a unified implementation for generating multi-view images from various conditions, facilitating applications such as text- and image-based 3D generation and texturing. We demonstrate

that MV-Adapter sets a new quality standard for multi-view image generation, and opens up new possibilities due to its adaptability and versatility.

# 1 INTRODUCTION

Multi-view image generation is a fundamental task with significant applications in areas such as 2D/3D content creation, robotics perception, and simulation. With the advent of text-to-image (T2I) diffusion models (Ramesh et al., 2022; Nichol et al., 2022; Saharia et al., 2022; Ramesh et al., 2021; Balaji et al., 2022; Podell et al., 2024; Mokady et al., 2023), there has been considerable progress in generating high-quality single-view images. Extending these models to handle multi-view generation holds the promise of unifying text, image, and 3D data into a cohesive framework.

Recent attempts on multi-view image generation (Shi et al., 2023b; Tang et al., 2023; 2024; Huang et al., 2024b; Gao et al., 2024; Liu et al., 2023a; Long et al., 2024; Li et al., 2024; Kant et al., 2024; Zheng & Vedaldi, 2024; Wang & Shi, 2023) involve fine-tuning T2I models on large-scale 3D datasets (Deitke et al., 2023; Yu et al., 2023) and propose modeling 3D consistency across images by applying self-attention on relevant pixels in different views. As a pioneer work, MVDream (Shi et al., 2023b) applies self-attention on latent pixels from all generated views, allowing the network to implicitly learn the consistency. Follow-up works like SPAD (Kant et al., 2024) and Era3D (Li et al., 2024) constrain the self-attention along epipolar lines, which improves efficiency and enables higher-resolution synthesis (Li et al., 2024).

While these advancements have led to progressively better results, they face several limitations that hinder their practicality. First, they often require full fine-tuning of pre-trained T2I models, which demands substantial computational resources and memory usage, making it impractical to scale to larger models and higher resolutions. The most advanced model to date is trained on Stable Diffusion 2-1 with 860M parameters at resolution 512 (Li et al., 2024). Second, full-parameter training with substantial network structure changes can lead to catastrophic forgetting of pre- trained knowledge, impairing compatibility with derivatives and extensions of the base model, including personalized models tailored to specific subjects or styles (Ruiz et al., 2023; Gal et al., 2022; Hu et al., 2021), distilled few-step models optimized for efficiency (Luo et al., 2023; Lin et al., 2024), and plugins (*e.g.* ControlNets (Zhang et al., 2023)) that add new functionalities. This incompatibility restricts the ability to leverage the continuous advancements and community contributions. Third, existing methods mainly serve a single purpose, for example generating multi-view images from text (Shi et al., 2023b; Kant et al., 2024), a reference image (Wang & Shi, 2023; Shi et al., 2023a; Wang et al., 2024b; Voleti et al., 2024; Wen et al., 2024; Li et al., 2024; Huang et al., 2024b), or geometry conditions (Bensadoun et al., 2024), but sharing the underlying logic of maintaining multi-view consistency. It is desirable to have a unified design that incorporates diverse conditioning signals, addressing the varied requirements of multi-view generation tasks across various domains.

To address these challenges, we propose MV-Adapter, a versatile plug-and-play adapter that enhances T2I models and their derivatives for multi-view generation under various conditions. Our approach eliminates the need for full model fine-tuning by introducing a multi-view adapter network seamlessly integrated with frozen T2Is. This significantly reduces computational costs and memory usage in training, making high-resolution generation feasible on larger models like Stable Diffusion XL (Podell et al., 2024). By preserving the original feature space of the base T2I model during training, MV-Adapter maintains high compatibility with various derivative models and community-developed plugins. This adaptability allows users to benefit from personalized subjects or styles, efficient few-step generation, and additional controllability without specific re-training. Moreover, we involve a unified design in the adapter network to support diverse conditioning inputs. It comprises a condition guider that processes camera or geometry guidance, enabling the model to incorporate viewpoint or structural information and therefore supports both 3D object generation and 3D model texture generation. This design also introduces decoupled attention blocks, which consists of multi-view attention layers and optional image cross-attention layers, allowing the model to generate from both text and image conditions.

We evaluate the performance of our MV-Adapter on a diverse set of personalized and efficient T2Is from the community. These models encompass a wide spectrum of domains, such as various styles and concepts, forming a comprehensive benchmark for our evaluation. Results of our experiments demonstrate promising outcomes.

In summary, contributions of MV-Adapter are as follows: (1) **Efficiency**. MV-Adapter eliminates the need for full fine-tuning, increasing training efficiency and enabling high-resolution generation. (2) **Adaptability**. MV-Adapter is fully compatible with derivatives and extensions of the base T2I model. (3) **Versatility**. MV-Adapter supports multiple conditioning inputs, broadening the scope of multi-view generation applications. (4) **Performance**. Experments demonstrate that T2Is with MV-Adapter can generate multi-view consistent images while preserving visual quality, leveraging the specific strengths of the base T2I models.

## 2 RELATED WORK

**Text-to-image diffusion models.**     Text-to-image (T2I) generation (Ramesh et al., 2022; Nichol et al., 2022; Saharia et al., 2022; Ramesh et al., 2021; Balaji et al., 2022; Podell et al., 2024; Mokady et al., 2023; Huang et al., 2024a) has made remarkable progress, particularly with the advancement of diffusion models (Ho et al., 2020; Song et al., 2020; Dhariwal & Nichol, 2021; Ho & Salimans, 2022). Guided diffusion (Dhariwal & Nichol, 2021) and classifier-free guidance (Ho & Salimans, 2022) improved text conditioning and generation fidelity. DALL-E2 (Ramesh et al., 2022) leverages CLIP (Radford et al., 2021) for better text-image alignment. The Latent Diffusion Model (Rombach et al., 2022), also known as Stable Diffusion, enhances efficiency by performing diffusion in the latent space of an autoencoder. Stable Diffusion XL (Podell et al., 2024), a two-stage cascade diffusion model, has greatly improved the generation of high-frequency details and overall image quality, elevating the aesthetic appeal of the outputs.

**Derivatives and extensions of T2I models.**     To facilitate creation with pre-trained T2Is, various derivative models and extensions have been developed, focusing on model distillation for efficiency (Meng et al., 2023; Song et al., 2023; Luo et al., 2023; Lin et al., 2024) and controllable generation (Cao et al., 2024). These derivatives encompass personalization (Ruiz et al., 2023; Gal et al., 2022; Hu et al., 2021; Shi et al., 2024; Wang et al., 2024a; Ma et al., 2024; Song et al., 2024; Kumari et al., 2023; Ye et al., 2023), and spatial control (Mou et al., 2024; Zhang et al., 2023). Typically, they employ adapters or fine-tuning methods to extend functionality while preserving the original feature space of the pre-trained models. For instance, DreamBooth (Ruiz et al., 2023) uses class-specific prior preservation loss for personalization, and ControlNet (Zhang et al., 2023) and T2I-Adapter (Mou et al., 2024) enable flexible control over generation by incorporating adapters to the base T2Is. Our work builds on these non-intrusive methods, ensuring compatibility with our MV-Adapter for broader applications.

**Multi-view Generation with T2I models.**     Multi-view generation methods (Shi et al., 2023b; Tang et al., 2023; 2024; Huang et al., 2024b; Gao et al., 2024; Liu et al., 2023a; Long et al., 2024; Li et al., 2024; Kant et al., 2024; Zheng & Vedaldi, 2024; Wang & Shi, 2023) extend T2I models by leveraging large-scale 3D datasets (Deitke et al., 2023; Yu et al., 2023). For instance, MVDream (Shi et al., 2023b) integrates camera embeddings and expands the self-attention mechanism from 2D to 3D for cross-view connections, while SPAD (Kant et al., 2024) enhances spatial relational modeling by applying epipolar constraints to cross-view attention. Era3D (Li et al., 2024) introduces an efficient row-wise self-attention mechanism aligned with epipolar lines across views, facilitating high-resolution multi-view generation. However, these methods typically require extensive parameter updates, altering the feature space of pre-trained T2I models and limiting their compatibility with T2I derivatives. Our work addresses this by introducing a multi-view adapter that harmonizes with pre-trained T2Is, significantly expanding the potential for diverse applications.

## 3 PRELIMINARY

Here we introduce the preliminary of multi-view diffusion models (Shi et al., 2023b; Kant et al., 2024; Li et al., 2024), which can help understand the common strategies in modeling multi-view consistency within T2I models.

**Multi-view diffusion models.**     Multi-view diffusion models enhance T2Is by introducing multi-view attention mechanism, enabling the generation of images that are consistent across different viewpoints. Several studies (Shi et al., 2023b; Wang & Shi, 2023) extend the self-attention of T2Is to include all pixels across multi-view images. Let $\boldsymbol{f}^{in}$ denotes the input of the attention block, the dense multi-view self-attention extends $\boldsymbol{f}^{in}$ from the view itself to the concatenated feature sequence

from $n$ views. While this approach captures global dependencies, it is computationally intensive, as it processes all pixels of all views. To mitigate the computational cost, epipolar attention (Kant et al., 2024; Huang et al., 2024b) leverages geometric relationships between views. Specifically, methods like SPAD (Kant et al., 2024) extend the self-attention by restricting $\boldsymbol{f}^{in}$ to the view itself as well as patches along its epipolar lines.

Furthermore, when generating orthographic views at an elevation angle of $0°$, the epipolar lines align with the image rows. Utilizing this property, row-wise self-attention (Li et al., 2024) is introduced after the original self-attention layers in T2I models. The process is defined as:

$$\boldsymbol{f}^{self} = \text{SelfAttn}(\boldsymbol{f}^{in}) + \boldsymbol{f}^{in}; \; \boldsymbol{f}^{mv} = \text{MultiViewAttn}(\boldsymbol{f}^{self}) + \boldsymbol{f}^{self} \tag{1}$$

where MultiViewAttn performs attention across the same rows in different views, effectively enforcing multi-view consistency with reduced computational overhead.

## 4 METHOD

MV-Adapter is a plug-and-play adapter that learns multi-view priors transferable to derivatives of T2Is without specific tuning, and enable them to generate multi-view consistent images under various conditions. As shown in Fig. 2, at inference, our MV-Adapter, which contains a condition guider and the decoupled attention layers, can be inserted into a personalized or distilled T2I to constitute the multi-view generator.

In detail, as shown in Fig. 3, the condition guider in Sec. 4.1 encodes the camera or geometry information, which supports both camera-guided and geometry-guided generation. Within the decoupled attention mechanism in Sec. 4.2, the additional multi-view attention layers learn multi-view consistency, while the optional image cross-attention layers are for image-conditioned generation. Sec. 4.3 elaborates on the training and inference processes of the MV-Adapter.

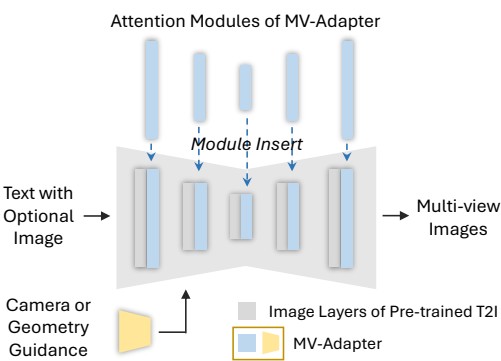

Figure 2: Inference pipeline.

### 4.1 CONDITION GUIDER

We design a general condition guider that supports encoding both camera and geometric representations, enabling T2I models to perform multi-view generation under various guidance.

**Camera conditioning.** MV-Adapter is designed for generating $n$ orthographic views. To condition on the camera pose, we use a camera ray representation ("raymap") that shares the same height and width as the latent representations and encodes the ray origin and direction at each spatial location (Watson et al., 2022; Sajjadi et al., 2022; Gao et al., 2024).

**Geometry conditioning.** Geometry-guided multi-view generation helps applications like texture generation. To condition on the geometry information, we use a global, rather than view-dependent representation that contains position maps and normal maps (Li et al., 2023; Bensadoun et al., 2024). Each pixel in the position map represents the coordinates of the point on the shape, which provide point correspondences across different views. Normal maps provide orientation information and capture fine geometric details, helping produce detailed textures. We concatenate the position map and normal map along to form a composite geometric conditioning input for each view.

**Encoder design.** To encode the camera or geometry representation, we design a simple and lightweight condition guider for the conditioning maps $\boldsymbol{c}_m$ ($\boldsymbol{c}_m \in \mathbb{R}^{n \times 6 \times h \times w}$). Inspired by T2I-Adapter (Mou et al., 2024), the condition guider consists of a series of convolutional networks, which contain feature extraction blocks and downsampling layers to adapt the feature resolution to the features in the U-Net encoder. The extracted multi-scale features are then added to the corresponding scales in the U-Net, enabling the model to integrate the conditioning information seamlessly at

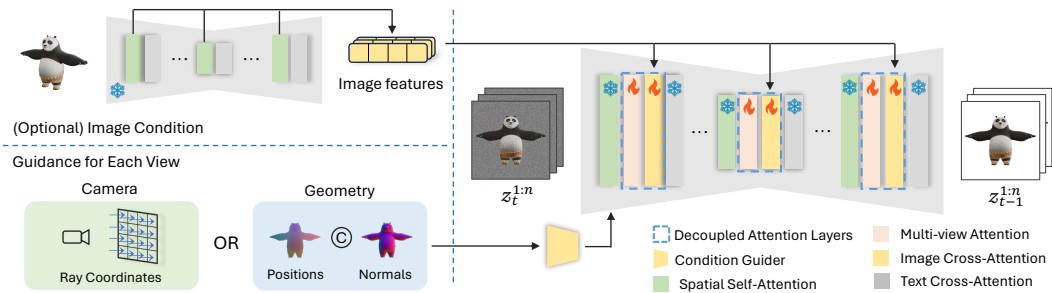

Figure 3: Overview of MV-Adapter. Our MV-Adapter consists of two components: 1) a condition guider that encodes camera or geometry condition; 2) decoupled attention layers that contain multi-view attention for learning multi-view consistency, and optional image cross-attention to support image-conditioned generation, where we use the pre-trained U-Net to encode fine-grained information of the reference image. After training, MV-Adapter can be inserted into any personalized or distilled T2I to generate multi-view images while leveraging the specific strengths of base models.

multiple levels. In theory, the input to our encoder is not limited to specific types of conditions; it can also be extended to a wider variety of maps, such as depth maps and pose maps.

### 4.2 DECOUPLED ATTENTION

We introduce a decoupled attention mechanism, where we retain the original spatial self-attention layers and add multi-view attention layers that enforce multi-view consistency as well as optional image cross-attention layers for image-conditioned generation. These three types of attention layers are organized in a parallel architecture, effectively leveraging the image priors from the pre-trained self-attention layers.

**Multi-view attention.** Considering the different applications of camera-guided and geometry-guided multi-view generation, we design different strategies for multi-view attention to meet the specific needs of each application (shown in Fig. 4(a)). For camera-guided generation, we follow Era3D (Li et al., 2024) to achieve image-to-3D creation, allowing the model to generate multi-view images at an elevation of $0°$. We then employ row-wise self-attention, restricting the multi-view attention to process only patches within the same row across views. For geometry-guided generation, considering the view coverage requirements of its main application (*i.e.*, texture generation), we adjust the distribution of the generated multi-view images. In addition to the four views evenly at elevation $0°$, we add two views from top and bottom. We perform both row-wise and column-wise self-attention, enabling efficient information exchange among all views.

**Image cross-attention.** To condition on reference images $c_i$ and achieve control over fine-grained appearance details, we propose a novel method for incorporating detailed information from the image without altering the original feature space of the T2I model. We employ the pre-trained T2I model itself as the image encoder. Specifically, we employ a frozen U-Net that is identical to the pre-trained SD U-Net (Rombach et al., 2022), with its weights initialized from the SD U-Net. During the feature extraction process, we pass the clear reference image into this frozen U-Net, setting the timestep $t = 0$, and then extract multi-scale features from the spatial self-attention layers. These fine-grained features contain detailed information about the subject and are injected into the denoising U-Net through the decoupled image cross-attention layers. In this way, we leverage the rich representations learned by the pre-trained model, enabling precise control over the generated content.

**Attention architecture.** In the pre-trained T2I model, the spatial self-attention layer and text cross-attention layer are connected serially through residual connections. Suppose feature sequence $f^{in}$ is the input of the attention block, we can express the process as

$$f^{self} = \text{SelfAttn}(f^{in}) + f^{in}; \ f^{cross} = \text{CrossAttn}(f^{self}) + f^{self} \tag{2}$$

A straightforward method to incorporate new attention layers is to append them after the original layers, connecting them in a serial manner. However, the sequential arrangement may not effectively

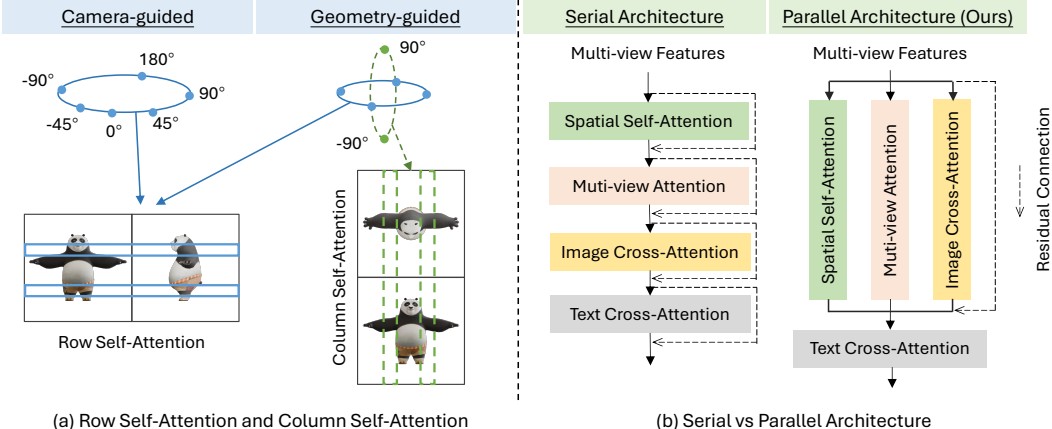

Figure 4: Overview of the decoupled attention design. (a) For camera-guided generation, similar to Era3D (Li et al., 2024), we apply row-wise self-attention to generate multi-view images at an elevations of $0°$. For geometry-guided generation, designed for texture generation, we add two views from the top and bottom to ensure comprehensive coverage and perform both row-wise and column-wise self-attention. (b) Instead of serially connecting new attention layers, which requires training additional modules from scratch, we utilize a parallel architecture that builds upon the established priors of pre-trained self-attention, enabling more efficient learning.

utilize the image priors modeled by the pre-trained self-attention layers, as it requires the new layers to learn from scratch. To fully exploit the effective priors of the spatial self-attention layers, we adopt a parallel architecture, as shown in Fig. 4(b). The process can be formulated as

$$\boldsymbol{f}^{self} = \text{SelfAttn}(\boldsymbol{f}^{in}) + \text{MultiViewAttn}(\boldsymbol{f}^{in}) + \text{ImageCrossAttn}(\boldsymbol{f}^{in}, \boldsymbol{f}^{ref}) + \boldsymbol{f}^{in} \quad (3)$$

where $\boldsymbol{f}^{ref}$ refers to features of the reference image. Since the features $\boldsymbol{f}^{in}$ fed into the new layers are the same as those to the self-attention layer, we can effectively initialize them with the pre-trained layers to transfer the image priors. We zero-initialize the output projection layer of the new layers to ensure that the initial output does not disrupt the original feature space. This architectural choice allows the model to build upon the established priors, facilitating efficient learning of multi-view consistency and image-conditioned generation, while preserving the original space of the base T2Is.

### 4.3 Training and Inference

During training, we only optimize the MV-Adapter, while freezing weights of the pre-trained T2I models. We train MV-Adapter on the dataset with pairs of a reference image, text and $n$ views, using the same training objective as T2I models:

$$\mathcal{L} = \mathbb{E}_{\mathcal{E}(\boldsymbol{x}_0^{1:n}), \boldsymbol{\epsilon} \sim \mathcal{N}(\boldsymbol{0}, \boldsymbol{I}), \boldsymbol{c}_t, \boldsymbol{c}_i, \boldsymbol{c}_m, t}[\|\boldsymbol{\epsilon} - \epsilon_\theta(\boldsymbol{z}_t^{1:n}, \boldsymbol{c}_t, \boldsymbol{c}_i, \boldsymbol{c}_m, t)\|_2^2] \quad (4)$$

where $\boldsymbol{c}_t$, $\boldsymbol{c}_i$ and $\boldsymbol{c}_m$ represent texts, reference images and conditioning maps (i.e., camera or geometry conditions) respectively. We randomly zero out the features of the reference image to drop image conditions, enabling classifier-free guidance at inference. Similar to prior work (Blattmann et al., 2023; Hoogeboom et al., 2023), we shift the noise schedule towards high noise levels as we move from the T2Is to the multi-view diffusion model that captures data of higher dimensionality. We shift the log signal-to-noise ratio by $\log(n)$, where $n$ is the number of generated views.

## 5 Experiments

We implemented MV-Adapter on Stable Diffusion V2.1 (SD2.1) and Stable Diffusion XL (SDXL), training a $512 \times 512$ adapter for SD2.1 and a $768 \times 768$ adapter for SDXL using a subset of the Objaverse dataset (Deitke et al., 2023). Detailed configurations are provided in the Appendix.

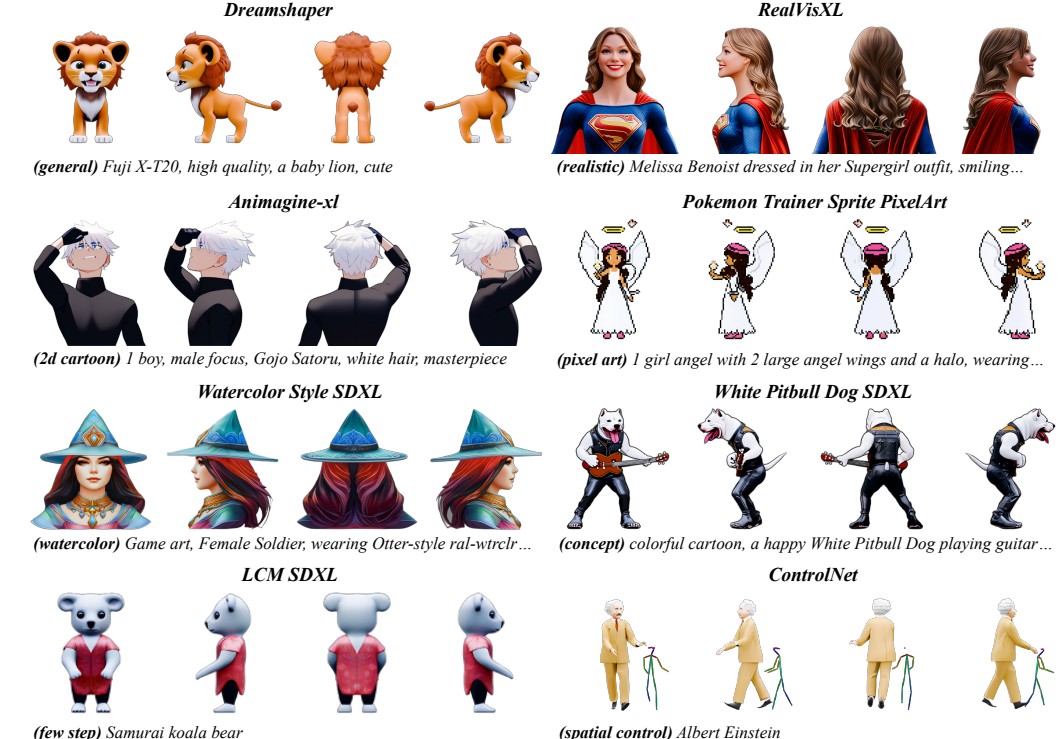

*Dreamshaper*

**(general)** *Fuji X-T20, high quality, a baby lion, cute*

*RealVisXL*

**(realistic)** *Melissa Benoist dressed in her Supergirl outfit, smiling...*

*Animagine-xl*

**(2d cartoon)** *1 boy, male focus, Gojo Satoru, white hair, masterpiece*

*Pokemon Trainer Sprite PixelArt*

**(pixel art)** *1 girl angel with 2 large angel wings and a halo, wearing...*

*Watercolor Style SDXL*

**(watercolor)** *Game art, Female Soldier, wearing Otter-style ral-wtrclr...*

*White Pitbull Dog SDXL*

**(concept)** *colorful cartoon, a happy White Pitbull Dog playing guitar...*

*LCM SDXL*

**(few step)** *Samurai koala bear*

*ControlNet*

**(spatial control)** *Albert Einstein*

Figure 5: Results with community models and extensions. Each sample corresponds to a distinct T2I model or extension. Information about the models can be found in the Appendix.

Table 1: Quantitative comparison on camera-guided text-to-multiview generation.

| Method | FID↓ | IS↑ | CLIP Score↑ |
|---|---|---|---|
| MVDream | 32.15 | 14.38 | 31.76 |
| SPAD | 48.79 | 12.04 | 30.87 |
| Ours (SD2.1) | 31.24 | 15.01 | 32.04 |
| Ours (SDXL) | **29.71** | **16.38** | **33.17** |

Table 2: Quantitative comparison on camera-guided image-to-multiview generation.

| Method | PSNR↑ | SSIM↑ | LPIPS↓ |
|---|---|---|---|
| ImageDream | 19.280 | 0.8472 | 0.1218 |
| Zero123++ | 20.312 | 0.8417 | 0.1205 |
| CRM | 20.185 | 0.8325 | 0.1247 |
| SV3D | 20.042 | 0.8267 | 0.1396 |
| Ouroboros3D | 20.810 | 0.8535 | 0.1193 |
| Era3D | 20.890 | 0.8601 | 0.1199 |
| Ours (SD2.1) | 20.867 | 0.8695 | 0.1147 |
| Ours (SDXL) | **22.131** | **0.8816** | **0.1002** |

## 5.1 CAMERA-GUIDED MULTI-VIEW GENERATION

**Evaluation on community models and extensions.** We evaluated MV-Adapter using representative T2Is and extensions, including personalized models (Ruiz et al., 2023; Hu et al., 2021), efficient distilled models (Luo et al., 2023; Lin et al., 2024), and plugins such as ControlNet (Zhang et al., 2023). We present eight qualitative results in Fig. 5. More results can be found in the Appendix.

**Comparison with baselines.** For text-to-multiview generation, we compared our MV-Adapter with MVDream (Shi et al., 2023b) and SPAD (Kant et al., 2024) on 1,000 prompts from the Objaverse dataset. The results are presented in Fig. 6 and Table 1. For image-to-multiview generation, we conduct comparison with ImageDream (Wang & Shi, 2023), Zero123++ (Shi et al., 2023a), CRM (Wang et al., 2024b), SV3D (Voleti et al., 2024), Ouroboros3D (Wen et al., 2024), and Era3D (Li et al., 2024) on the Google Scanned Objects (GSO) dataset (Downs et al., 2022), as results shown in Fig. 7 and Table 2. Experiments indicate that, by preserving the original feature space of T2I models, our MV-Adapter achieves higher visual fidelity and consistency with conditions.

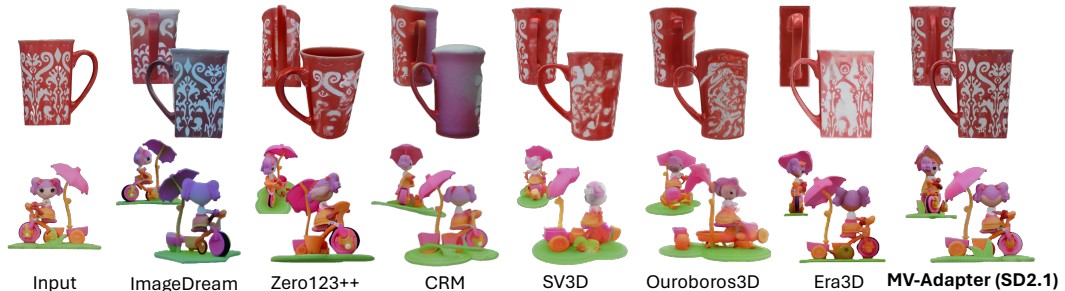

*Corgi **riding** a rocket*

*A character in blue and white armor*

| Input | MVDream | SPAD | **MV-Adapter (SD2.1)** | **MV-Adapter (SDXL)** |

Figure 6: Qualitative comparison on camera-guided text-to-multiview generation. our MV-Adapter achieves higher visual fidelity and image-text consistency.

| Input | ImageDream | Zero123++ | CRM | SV3D | Ouroboros3D | Era3D | **MV-Adapter (SD2.1)** |

Figure 7: Qualitative comparison on camera-guided image-to-multiview generation.

## 5.2 GEOMETRY-GUIDED MULTI-VIEW GENERATION

**Evaluation on community models and extensions.** We evaluated our geometry-guided model with T2I derivative models. The results in Fig. 8 demonstrate the adaptability of MV-Adapter in seamlessly integrating with different base models.

**Comparison with baselines.** We compare our text- and image-conditioned multi-view-based texture generation method (see Sec. 5.4) with four state-of-the-art methods, including TEXTure (Richardson et al., 2023), Text2Tex (Chen et al., 2023), Paint3D (Zeng et al., 2024), SyncMVD (Liu et al., 2023b), and FlashTex (Deng et al., 2024). For our image-to-texture model, we used ControlNet (Zhang et al., 2023) to generate reference images conditioned on text and depth maps. As shown in Fig. 10 and Table 3, compared to these project-and-inpaint or synchronized multi-view texturing methods, our approach fine-tunes additional modules to model geometric associations and preserves the generative capabilities of the base T2I model, thereby producing multi-view consistent and high-quality textures. Additionally, testing on a single RTX 4090 GPU revealed that our method achieves faster generation speeds than the others.

Table 3: Quantitative comparison on 3D texture generation. FID and KID ($\times 10^{-4}$) are evaluated on multi-view renderings. Our models achieves best texture quality with faster inference.

| Method | FID↓ | KID↓ | Time↓ |
|---|---|---|---|
| TEXTure | 56.44 | 61.16 | 90s |
| Text2Tex | 58.43 | 60.81 | 421s |
| Paint3D | 44.38 | 47.06 | 60s |
| SyncMVD | 36.13 | 42.28 | 50s |
| FlashTex | 50.48 | 56.36 | 186s |
| Ours (SD2.1 - Text) | 38.19 | 42.83 | **18s** |
| Ours (SD2.1 - Image) | 33.93 | 38.73 | 19s |
| Ours (SDXL - Text) | 32.75 | 35.18 | 32s |
| Ours (SDXL - Image) | **27.28** | **29.47** | 33s |

## 5.3 ABLATION STUDY

We conduct ablation studies to evaluate the efficiency and adaptability of our MV-Adapter, as well as the detailed design of the adapter network.

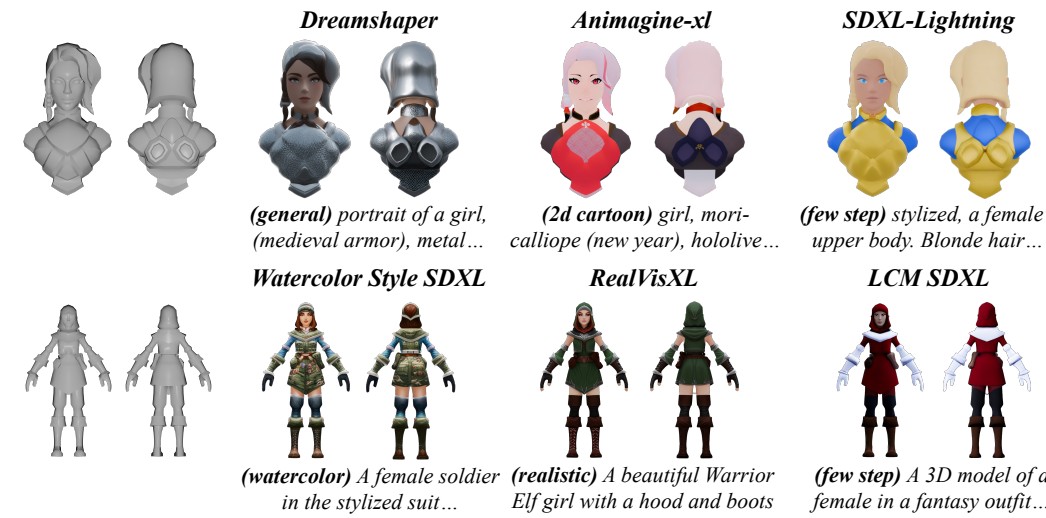

***Dreamshaper***

***Animagine-xl***

***SDXL-Lightning***

*(general) portrait of a girl, (medieval armor), metal...*

*(2d cartoon) girl, mori-calliope (new year), hololive...*

*(few step) stylized, a female's upper body. Blonde hair...*

***Watercolor Style SDXL***

***RealVisXL***

***LCM SDXL***

*(watercolor) A female soldier in the stylized suit...*

*(realistic) A beautiful Warrior Elf girl with a hood and boots*

*(few step) A 3D model of a female in a fantasy outfit...*

Figure 8: Results of geometry-guided text-to-multiview generation with community models.

**Efficiency.** To assess the training efficiency of our adapter design, we conducted comparison with Era3D (Li et al., 2024), which requires full training rather than fine-tuning only adapters like us. We further extend this model to SDXL (Podell et al., 2024) for a comprehensive evaluation. As shown in Table 4, our MV-Adapter significantly reduces training costs, facilitating high-resolution multi-view generation based on larger backbones.

Table 4: Comparison of training costs with full-tuning methods (batch size set to 1).

| Method | Trainable params ↓ | Memory usage↓ | Training speed ↑ |
|---|---|---|---|
| Era3D (SD2.1) | 993M | 36G | 2.2iter/s |
| Ours (SD2.1) | **127M** | **17G** | **3.1iter/s** |
| Era3D (SDXL) | 3.1B | >80G | - |
| Ours (SDXL) | **490M** | **60G** | **1.05iter/s** |

**Adaptability.** We compare MV-Adapter with the full-trained text-to-multiview generation method MVDream (Shi et al., 2023b) regarding compatibility with T2I derivatives. MVDream, which fine-tunes the whole T2I model, cannot be easily replaced with other T2Is; thus, we integrate LoRA (Hu et al., 2021) for our experiments. As shown in Fig. 9, MVDream struggles to generate images that align with the text and style, whereas our MV-Adapter produces high-quality results, demonstrating its superior adaptability.

*(3d style) 1 girl, blue eyes, upper body, mask, eyes half closed*

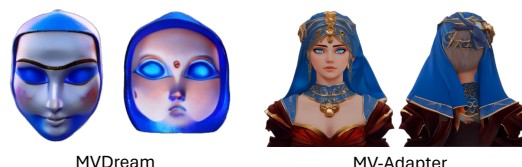

MVDream          MV-Adapter

Figure 9: Qualitative ablation study on the adaptability of MV-Adapter.

**Network design.** We conducted ablation studies on our proposed image encoder and parallel attention architecture. Specifically, we compare the settings of a) using CLIP (Radford et al., 2021) for encoding reference images instead of SD U-Net, and b) replacing the parallel architecture with a serial counterpart, with

Table 5: Quantitative ablation studies on attention architecture.

| Method | PSNR↑ | SSIM↑ | LPIPS↓ |
|---|---|---|---|
| Serial (SDXL) | 20.687 | 0.8681 | 0.1149 |
| Parallel (SDXL) | **22.131** | **0.8816** | **0.1002** |

c) our MV-Adapter. As shown in Fig. 11, comparing a) and c) reveals that CLIP capture only coarse, semantic-level information, while the pre-trained U-Net encodes finer details, producing results closely aligned with the input. Comparing b) and c) shows that, the serial setting, which does not leverage the pre-trained image prior, tends to produce artifacts and misaligned details. Our MV-Adapter achieves greater consistency both among generated views and with the reference image, especially at the detail level. More results can be found in the Appendix.

*A gray raccoon 3D model with a long tail, pointy ears, black eyes, and a pink nose.*

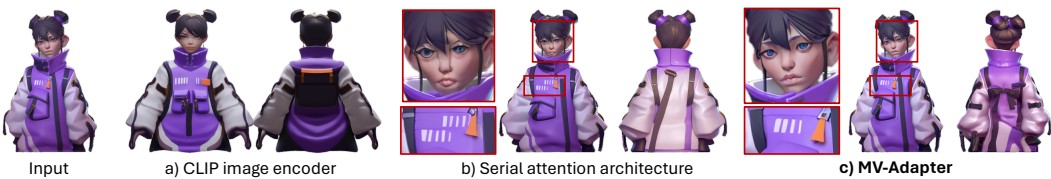

Figure 10: Qualitative comparison on texture generation. We compare our text- and image-conditioned models with baseline methods.

Figure 11: Qualitative ablation study on the network design.

### 5.4 APPLICATIONS

**3D generation.** We follow the existing pipelines (Li et al., 2024; Wu et al., 2024) to achieve 3D generation. After generating multi-view images from text or image conditions using MV-Adapter, we use StableNormal (Ye et al., 2024) to generate corresponding normal maps. The multi-view images and normal maps are then fed into NeuS (Wang et al., 2021) to reconstruct the 3D mesh. The generated results are shown in the Appendix.

**Texture generation.** We use backprojection and incidence-based weighted blending techniques (Bensadoun et al., 2024) to map the generated multi-view images onto the UV texture map. Despite optimizing view distribution to enhance coverage, some areas may remain uncovered due to occlusions or extreme angles. To address this, we perform view coverage analysis to identify uncovered regions, render images from the current 3D texture for those views, and refine them using an efficient inpainting model (Suvorov et al., 2022). We show more visual results in the Appendix.

## 6 CONCLUSION

In this paper, we introduce MV-Adapter, a versitile plug-and-play adapter that enhances text-to-image (T2I) diffusion models and their derivatives for multi-view generation under various conditions, without compromising quality or modifying the original feature space. Our approach incorporates a condition guider and a decoupled attention mechanism, enabling both camera-guided and geometry-guided multi-view generation from text and images. Once trained, our MV-Adapter can be seamlessly integrated into various T2I models—including personalized, distilled, and plugin-enhanced models—to generate multi-view images with high consistency and visual fidelity. Extensive evaluations highlight the efficiency, adaptability, and versatility of MV-Adapter across different models and conditions. Furthermore, we extend our multi-view generation framework to support applications such as 3D generation and texture generation. Overall, MV-Adapter offers an efficient and flexible solution for multi-view image generation, significantly broadening the capabilities of pre-trained T2I models and presenting exciting possibilities for a wide range of applications.

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

# A APPENDIX

## A.1 BACKGROUND

**Stable Diffusion (SD) and Stable Diffusion XL (SDXL).** We adopt Stable Diffusion (Rombach et al., 2022) and Stable Diffusion XL (Podell et al., 2024) as our base T2I models, since they have a well-developed community with many powerful derivatives for evaluation. SD and SDXL perform the diffusion process within the latent space of a pre-trained autoencoder $\mathcal{E}(\cdot)$ and $\mathcal{D}(\cdot)$. In training, an encoded image $z_0 = \mathcal{E}(x_0)$ is perturbed to $z_t$ at step $t$ by the forward diffusion. The denoising network $\epsilon_\theta$ learns to reverse this process by predicting the added noise, encouraged by an MSE loss:

$$\mathcal{L} = \mathbb{E}_{\mathcal{E}(x_0), \epsilon \sim \mathcal{N}(0, I), c, t}[\|\epsilon - \epsilon_\theta(z_t, c, t)\|_2^2] \tag{5}$$

where $c$ denotes the conditioning texts. In SD, $\epsilon_\theta$ is implemented as a UNet (Ronneberger et al., 2015) consisting of pairs of down/up sample blocks and a middle block. Each block contains pairs of spatial self-attention layers and cross-attention layers, which are serially connected using the residual structure. SDXL leverages a three times larger UNet backbone than SD for high-resolution image synthesis, and introduces a refinement denoiser to improve the visual fidelity.

## A.2 IMPLEMENTATION DETAILS

**Dataset.** We trained MV-Adapter on a filtered high-quality subset of the Objaverse dataset (Deitke et al., 2023), comprising approximately 70,000 samples, with captions from Cap3D (Luo et al., 2024). To accommodate the efficient multi-view self-attention mechanism, we rendered orthographic views to train the the model to generate $n = 6$ views per sample. For the camera-guided

generation, we rendered views of 3D models with the elevation angle set to $0°$ and azimuth angles at $\{0°, 45°, 90°, 180°, 270°, 315°\}$. This distribution aligns with the setting used in Era3D (Li et al., 2024), facilitating the application of a similar image-to-3D pipeline for 3D generation tasks. For the geometry-guided generation, we included four views at an elevation of $0°$ with azimuth angles of $\{0°, 90°, 180°, 270°\}$, added two additional views from the top and bottom. In addition to the target views, we rendered five random views within a certain frontal range of the models to serve as reference images during training.

**Training.** We utilized two versions of Stable Diffusion (Rombach et al., 2022) as the base models for training. Specifically, we trained a 512-resolution model based on Stable Diffusion 2.1 (SD2.1) and a 768-resolution model based on Stable Diffusion XL (SDXL). During training, we randomly dropped the text condition with a probability of 0.1, the image condition with a probability of 0.1, and both text and image conditions simultaneously with a probability of 0.1. Following prior work (Hoogeboom et al., 2023; Blattmann et al., 2023), we shifted the noise schedule to higher noise levels by adjusting the log signal-to-noise ratio (SNR) by $\log(n)$, where $n = 6$ is the number of the generated views. For the specific training configurations, we used a learning rate of $5 \times 10^{-5}$ and trained the MV-Adapter on 8 NVIDIA A100 GPUs for 10 epochs.

**Inference.** In our experimental setup, we generated multi-view images using the DDPM sampler (Ho et al., 2020) with classifier-free guidance (Ho & Salimans, 2022), and set the number of inference steps to 50. For generation conditioned solely on text (i.e., setting the weight of the image condition $\lambda_i$ to 0), we set the guidance scale to 7.0. For image-conditioned generation, we set the guidance scale of image condition $\alpha$ and text condition $\beta$ to 3.0. Following TOSS (Shi et al., 2023c), the calculation can be expressed as:

$$\hat{\epsilon}_\theta(\boldsymbol{z}_t^{1:n}, \boldsymbol{c}_t, \boldsymbol{c}_i, \boldsymbol{c}_m, t) = \epsilon_\theta(\boldsymbol{z}_t^{1:n}, \emptyset, \emptyset, \boldsymbol{c}_m, t)$$
$$+ \alpha \left[ \epsilon_\theta(\boldsymbol{z}_t^{1:n}, \emptyset, \boldsymbol{c}_i, \boldsymbol{c}_m, t) - \epsilon_\theta(\boldsymbol{z}_t^{1:n}, \emptyset, \emptyset, \boldsymbol{c}_m, t) \right]$$
$$+ \beta \left[ \epsilon_\theta(\boldsymbol{z}_t^{1:n}, \boldsymbol{c}_t, \boldsymbol{c}_i, \boldsymbol{c}_m, t) - \epsilon_\theta(\boldsymbol{z}_t^{1:n}, \emptyset, \boldsymbol{c}_i, \boldsymbol{c}_m, t) \right] \quad (6)$$

where $\boldsymbol{c}_t$, $\boldsymbol{c}_i$ and $\boldsymbol{c}_m$ represent texts, reference images and conditioning maps (*i.e.*, camera or geometry conditions) respectively. Since we did not drop $\boldsymbol{c}_m$ during the training process, we do not use the classifier-free guidance method for it.

**Comparison with baselines.** We conducted comprehensive comparisons with baseline methods across three settings: text-to-multiview generation, image-to-multiview generation, and texture generation. In these experiments, we evaluated both versions of MV-Adapter based on Stable Diffusion 2.1 (SD2.1) (Rombach et al., 2022) and Stable Diffusion XL (SDXL) (Podell et al., 2024), demonstrating the performance gains brought by MV-Adapter due to its efficient training and scalability.

For text-to-multiview generation, we selected MVDream (Shi et al., 2023b) and SPAD (Kant et al., 2024) as baseline methods. MVDream extends the original self-attention mechanism of T2I models to the multi-view domain. SPAD introduces epipolar constraints into the multi-view attention mechanism. We tested on 1,000 prompts selected from the Objaverse dataset (Deitke et al., 2023). We computed Fréchet Inception Distance (FID), Inception Score (IS), and CLIP Score on all generated views to assess the quality of the generated images and their alignment with the textual prompts.

For image-to-multiview generation, we compared our method with ImageDream (Wang & Shi, 2023), Zero123++(Shi et al., 2023a), CRM(Wang et al., 2024b), SV3D (Voleti et al., 2024), Ouroboros3D (Wen et al., 2024), and Era3D (Li et al., 2024). ImageDream, Zero123++, CRM, and Era3D generally fall into the category of modifying the original network architecture of T2I models to extend them for multi-view generation. SV3D and Ouroboros3D fine-tune text-to-video (T2V) models to achieve multi-view generation. We selected 100 assets covering multiple object categories from the Google Scanned Objects (GSO) dataset (Downs et al., 2022) as our test set. For each asset, we rendered input images from front-facing views, with input views randomly distributed in azimuth angles between $-45°$ and $45°$ and elevation angles between $-10°$ and $30°$. We evaluated the generated multi-view images by computing Peak Signal-to-Noise Ratio (PSNR), Structural Similarity Index Measure (SSIM), and Learned Perceptual Image Patch Similarity (LPIPS) between the generated images and the ground truth, assessing both the consistency and quality of the outputs.

For 3D texture generation, we compared our text-based and image-based models with project-and-paint methods such as TEXTure (Richardson et al., 2023), Text2Tex (Chen et al., 2023), and

Table 6: Community models and extensions for evaluation.

| Category | Model Name | Domain | Model Type |
|---|---|---|---|
| Personalized T2I | Dreamshaper[1] | General | T2I Base Model |
| | RealVisXL[2] | Realistic | T2I Base Model |
| | Animagine-xl[3] | 2D Cartoon | T2I Base Model |
| | 3D Render Style XL[4] | 3D Cartoon | LoRA |
| | Pokemon Trainer Sprite PixelArt[5] | Pixel Art | LoRA |
| | Chalk Sketch SDXL[6] | Chalk Sketch | LoRA |
| | Chinese Ink LoRA[7] | Color Ink | LoRA |
| | Zen Ink Wash Sumi-e[8] | Wash Ink | LoRA |
| | Watercolor Style SDXL[9] | Watercolor | LoRA |
| | Papercut SDXL[10] | Papercut | LoRA |
| | Furry Enhancer[11] | Enhancer | LoRA |
| | White Pitbull Dog SDXL[12] | Concept | LoRA |
| | Spider spirit fourth sister[13] | Concept | LoRA |
| Distilled T2I | SDXL-Lightning[14] | Few Step | T2I Base Model |
| | LCM-SDXL[15] | Few Step | T2I Base Model |
| Extension | ControlNet Openpose[16] | Spatial Control | Plugin |
| | ControlNet Scribble[17] | Spatial Control | Plugin |
| | ControlNet Tile[18] | Image Deblur | Plugin |
| | T2I-Adapter Sketch[19] | Spatial Control | Plugin |
| | IP-Adapter[20] | Image Prompt | Plugin |

Paint3D (Zeng et al., 2024), the synchronized multi-view texturing method SyncMVD (Liu et al., 2023b), and the optimization-based method FlashTex (Deng et al., 2024). We randomly selected 200 models along with their captions from the Objaverse (Deitke et al., 2023) dataset for testing. Multiple views were rendered from the generated 3D textures, and we computed FID and Kernel Inception Distance (KID) of them to evaluate the quality of the generated textures. Additionally, we recorded the texture generation time to assess the inference efficiency of each method.

**Community models and extensions for evaluation.** To ensure a comprehensive benchmark, we selected a diverse set of representative T2I derivative models and extensions from the community

---

[1]https://civitai.com/models/112902?modelVersionId=126688

[2]https://civitai.com/models/139562?modelVersionId=789646

[3]https://huggingface.co/cagliostrolab/animagine-xl-3.1

[4]https://huggingface.co/goofyai/3d_render_style_xl

[5]https://civitai.com/models/159333/pokemon-trainer-sprite-pixelart?modelVersionId=443092

[6]https://huggingface.co/JerryOrbachJr/Chalk-Sketch-SDXL

[7]https://huggingface.co/ming-yang/sdxl_chinese_ink_lora

[8]https://civitai.com/models/647926/zen-ink-wash-sumi-e-sdxl-pony-flux?modelVersionId=724876

[9]https://civitai.com/models/484723/watercolor-style-sdxl

[10]https://huggingface.co/TheLastBen/Papercut_SDXL

[11]https://civitai.com/models/310964/furry-enhancer?modelVersionId=558568

[12]https://civitai.com/models/700883/white-pitbull-dog-sdxl?modelVersionId=787948

[13]https://civitai.com/models/689010/pony-black-myth-wukong-spider-spirit-fourth-sister?modelVersionId=771146

[14]https://huggingface.co/ByteDance/SDXL-Lightning

[15]https://huggingface.co/latent-consistency/lcm-sdxl

[16]https://huggingface.co/xinsir/controlnet-openpose-sdxl-1.0

[17]https://huggingface.co/xinsir/controlnet-scribble-sdxl-1.0

[18]https://huggingface.co/xinsir/controlnet-tile-sdxl-1.0

[19]https://huggingface.co/TencentARC/t2i-adapter-sketch-sdxl-1.0

[20]https://huggingface.co/h94/IP-Adapter

for evaluation. As illustrated in Table 6, these models include personalized models that encompass various domains such as anime, stylistic paintings, and realistic photographic images, as well as efficient distilled models and plugins for controllable generation. They cover a wide range of subjects, including portraits, animals, landscapes, and more. This selection enables a thorough evaluation of our approach across different styles and content, demonstrating the adaptability and generality of MV-Adapter in working with various T2I derivatives and extensions.

### A.3 Additional Discussions

#### A.3.1 MV-Adapter vs. Multi-view LoRA

LoRA (Low-Rank Adaptation) (Hu et al., 2021) offers an alternative approach to achieving plug-and-play multi-view generation. Specifically, using a condition encoder to inject camera representations, we extend the original self-attention mechanism to operate across all pixels of multiple views. During training, we introduce trainable LoRA layers into the network, allowing these layers to learn multi-view consistency or, optionally, generate images conditioned on a reference view. This approach requires the spatial self-attention mechanism to simultaneously capture spatial image knowledge, ensure multi-view consistency, and align generated images with reference views.

However, the multi-view LoRA approach has a notable limitation. The "incremental changes" it introduces to the network are **not orthogonal or decoupled** from those induced by T2I derivatives, such as personalized T2I models or LoRAs. Specifically, layers fine-tuned by multi-view LoRA and those tuned by personalized LoRA often overlap. Note that each weight matrix learned by both represents a linear transformation defined by its columns, so it is intuitive that the merger would retain the information available in these columns only when the columns that are being added are orthogonal to each other (Shah et al., 2023). Clearly, the multi-view LoRA and personalized models are not orthogonal, which often leads to challenges in retaining both sets of learned knowledge. This can result in a trade-off where either multi-view consistency or the fidelity of concepts (such as style or subject identity) is compromised.

In contrast, our proposed **decoupled** attention mechanism encourages different attention layers to specialize in their respective tasks without needing to fine-tune the original spatial self-attention layers. In this design, the layers we train do not overlap with those in the original T2I model, thereby better preserving the original feature space and enhancing compatibility with other models.

We conducted a series of experiments to test these approaches. We trained two versions of multi-view LoRA, targeting different modules: (1) inserting LoRA layers only into the attention layers, and (2) inserting LoRA layers into multiple layers, including the convolutional layers, down-sampling, up-sampling layers, etc. For both settings, we set the LoRA rank to 64 and alpha to 32. As shown in Fig. 12 and Fig. 13, while the multi-view LoRA approach can generate multi-view consistent images when the base model is not changed, it often struggles to maintain multi-view consistency when switching to a different base model or when integrating a new LoRA. In contrast, as demonstrated in Fig. 14, our MV-Adapter, equipped with the decoupled attention mechanism, maintains consistent multi-view generation even when used with personalized models.

Compared to the LoRA mechanism, our decoupled attention-based approach proves more robust and adaptable for extending T2I models to multi-view generation, offering greater flexibility and compatibility with various pre-trained models.

#### A.3.2 Adaptability of Image-conditioned Model

Evaluating the adaptability of the image-conditioned MV-Adapter on personalized models poses a challenge because the reference image already provides detailed subject-specific appearance guidance for multi-view generation. As a result, it's difficult to assess how well the model adapts when the subject's details are pre-defined. To address this, we conducted experiments on efficient distilled models, such as SDXL-Lightning (Lin et al., 2024). As illustrated in Fig. 15, after replacing the base model with a distilled T2I variant, the MV-Adapter was able to generate high-quality and multi-view consistent images **in just four steps**.

*(Base model: SDXL) Daenerys Targaryen from game of throne, full body, blender 3d, art station*

*(Base model: Animagine-xl) 1 girl, pink hair, pink shirts, smile, shy, masterpiece*

*(LoRA: Watercolor Style) painting, Burmese Cat, wearing ral-wtrclr, Comic book art*

Figure 12: Results of multi-view LoRA (set target modules to attention layers). The azimuth angles of the images from left to right are $0°, 45°, 90°, 180°, 270°, 315°$, corresponding to the front, front-left, left, back, right, and front-right of the object.

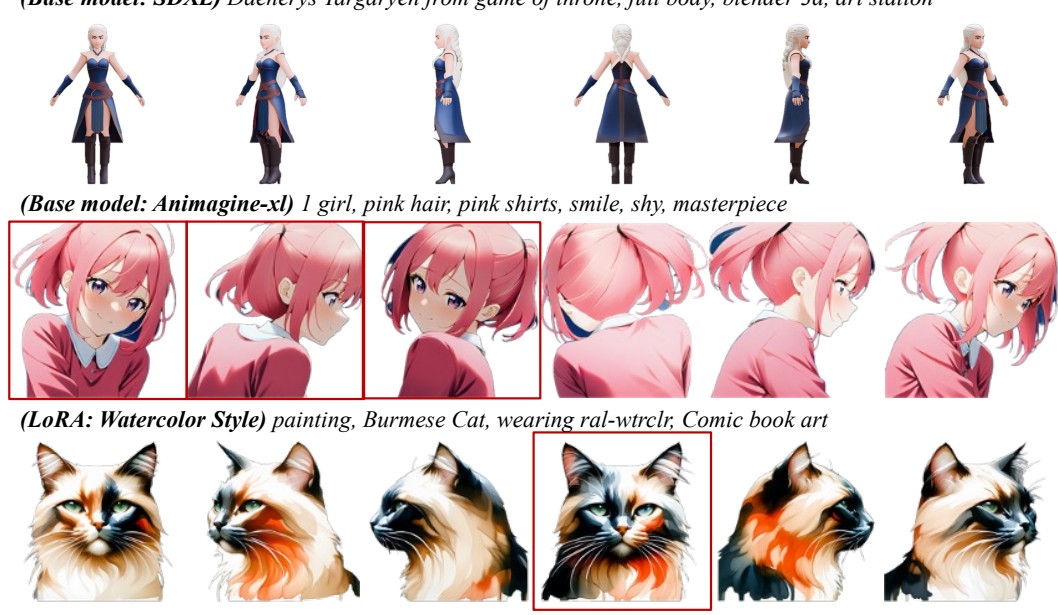

*(Base model: SDXL) Daenerys Targaryen from game of throne, full body, blender 3d, art station*

*(Base model: Animagine-xl) 1 girl, pink hair, pink shirts, smile, shy, masterpiece*

*(LoRA: Watercolor Style) painting, Burmese Cat, wearing ral-wtrclr, Comic book art*

Figure 13: Results of multi-view LoRA (set target modules to attention layers, convolutional layers, etc.). The azimuth angles of the images from left to right are $0°, 45°, 90°, 180°, 270°, 315°$, corresponding to the front, front-left, left, back, right, and front-right of the object.

The experiments clearly demonstrate that our image-conditioned MV-Adapter exhibits strong adaptability. Even when integrated into distilled models, it is capable of rapidly generating high-quality multi-view images, proving its efficiency and versatility.

*(Base model: SDXL) Daenerys Targaryen from game of throne, full body, blender 3d, art station*

*(Base model: Animagine-xl) 1 girl, pink hair, pink shirts, smile, shy, masterpiece*

*(LoRA: Watercolor Style) painting, Burmese Cat, wearing ral-wtrclr, Comic book art*

Figure 14: Results of MV-Adapter, which introduces decoupled attention mechanism rather than LoRA. The azimuth angles of the images from left to right are $0°, 45°, 90°, 180°, 270°, 315°$, corresponding to the front, front-left, left, back, right, and front-right of the object.

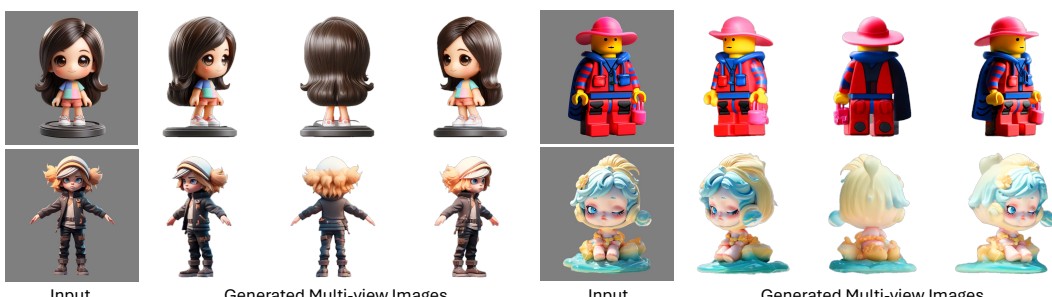

Input          Generated Multi-view Images          Input          Generated Multi-view Images

Figure 15: Results of MV-Adapter on camera-guided image-to-multiview generation with SDXL-Lightning (Lin et al., 2024) (number of inference steps set to 4).

### A.3.3 IMAGE RESTORATION CAPABILITIES

During the training of MV-Adapter, we probabilistically compress the resolution of reference images in the training data pairs to enhance the robustness of multi-view generation from images. We observed that the model trained with this approach is capable of generating high-resolution, detailed multi-view images even when the input is low-resolution, as depicted in Fig. 16. Through such training strategy, MV-Adapter has inherent image restoration capabilities and automatically enhances and refines input images during the generation process.

### A.3.4 SERIAL VS. PARALLEL ATTENTION ARCHITECTURE

To assess the effectiveness of our proposed parallel attention architecture, we conducted ablation studies on image-to-multi-view generation setting. As shown in Fig. 17, the serial setting, which cannot leverage the pre-trained image prior, tends to produce artifacts and inconsistent details with the image input. In contrast, our parallel setting produces high-quality and highly consistent results.

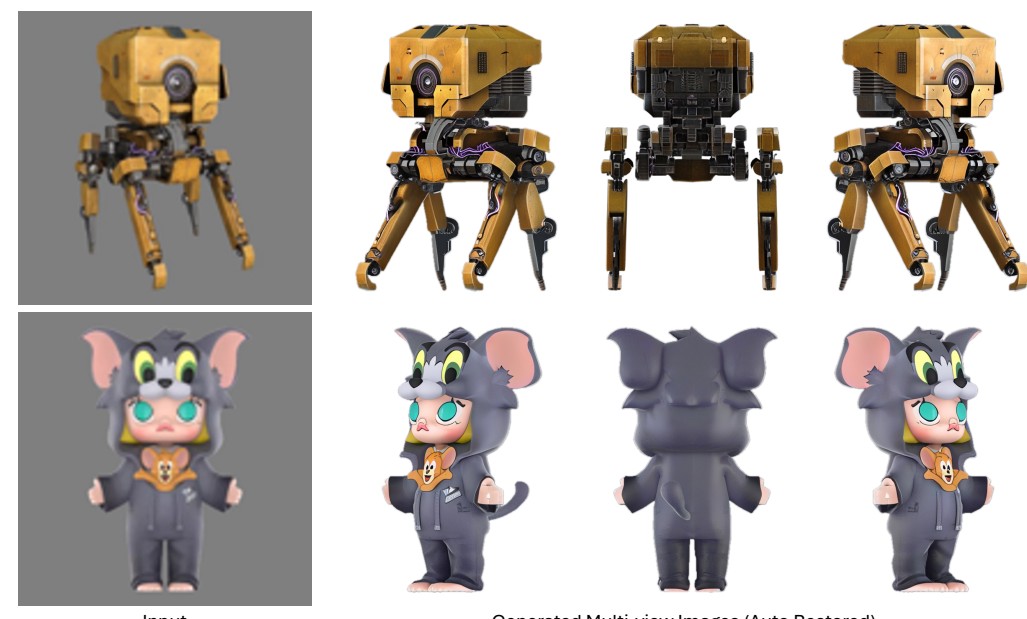

Input                  Generated Multi-view Images (Auto Restored)

Figure 16: Results on camera-guided image-to-multiview generation with low-resolution images as input.

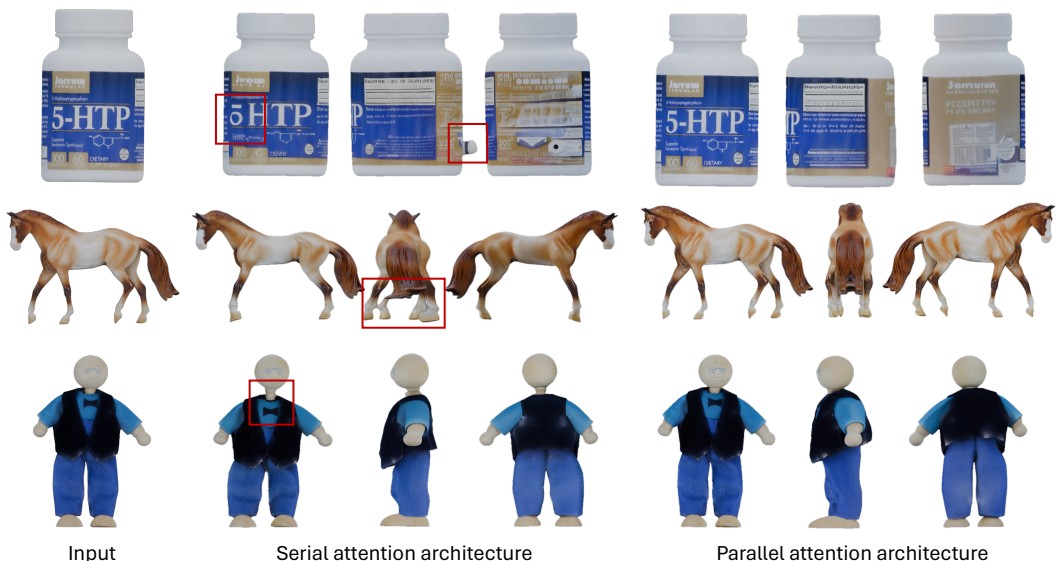

Input        Serial attention architecture          Parallel attention architecture

Figure 17: Qualitative ablation study on the attention architecture.

### A.3.5 APPLICABILITY OF MV-ADAPTER

**Broader potential applications.** Beyond the demonstrated applications in 3D object generation and 3D texture mapping, the MV-Adapter's strong adaptability and versatility open up a wide array of potential uses in image creation and personalization. For instance, creators can integrate MV-Adapter with their personalized T2I models—customized for specific identities or artistic styles—to generate multi-view images that capture consistent perspectives of their unique concepts. Additionally, MV-Adapter can facilitate tasks like multi-view portrait generation, where a subject's face is rendered consistently across different angles, or stylized multi-view illustrations that maintain artistic coherence across diverse perspectives.

**Inspiration for related tasks.** Our MV-Adapter represents a successful practice of decoupling image priors from geometric knowledge within T2I diffusion models. This approach provides valuable insights for downstream tasks that rely on image priors but also require modeling of geometric, physical, or temporal aspects. Specifically, characteristics related to geometry and viewpoint—such as zooming in/out, lighting variations, and shadow dynamics—can potentially be addressed by introducing new layers that decouple these factors or by fine-tuning the multi-view attention layers of MV-Adapter. By extending this decoupled architecture, it may be possible to model geometric-related properties more effectively, enabling advancements in areas like view-dependent appearance synthesis, relighting, and even animation where temporal consistency is crucial. This opens avenues for future research to explore how similar strategies can be applied to disentangle and control other complex factors in image generation tasks.

### A.3.6 Extending MV-Adapter for Arbitrary View Synthesis

In the main text, we introduced a novel adapter architecture—comprising parallel attention layers and a unified condition encoder—to achieve multi-view generation. We implemented efficient row-wise and column-wise attention mechanisms tailored for two specific applications: 3D object generation and 3D texture mapping, generating six views accordingly. However, our adapter framework is not limited to these configurations and can be extended to perform arbitrary view synthesis. To explore this capability, we designed a corresponding approach and conducted experiments, training a new version of MV-Adapter to handle arbitrary viewpoints.

Following CAT3D (Gao et al., 2024), we perform multiple rounds of multi-view generation, with the number of views generated each time set to $n = 8$. Starting from text or an initial single image as input, we first generate eight anchor views that broadly cover the object. In practice, these anchor views are positioned at elevations of $0°$ and $30°$, with azimuth angles evenly distributed around the circle (*e.g.* every $45°$). For generating new target views, we cluster the viewpoints based on their spatial orientations, grouping them into clusters of $8$. We then select the $4$ nearest known views from the already generated anchor views to serve as conditions guiding the generation of each target view.

In terms of implementation, the overall framework of our MV-Adapter remains unchanged. We adjust its inputs and specific attention components to accommodate arbitrary view synthesis. First, we set the number of input images to either $1$ or $4$. When using four input views, we concatenate them into a long image and input this into the pre-trained T2I U-Net to extract features. This simple yet effective method allows the images from the four views to interact within the pre-trained U-Net without requiring additional camera embeddings to represent these views. Second, we utilize full self-attention in the multi-view attention component, expanding the attention scope to enable the generation of target views with more flexible distributions.

To train an MV-Adapter capable of generating arbitrary viewpoints, we rendered data from 40 different views, with elevations of $-10°, 0°, 10°, 20°, 30°$, and azimuth angles evenly distributed around 360 degrees at each elevation layer. We trained the model for 16 epochs. During the first 8 epochs, the model was trained using a setting of one conditional view and eight target anchor views. In the subsequent 8 epochs, we trained with an equal mixture of one condition plus eight target views and four conditions plus eight target views.

As shown in Fig. 18, the visualization results demonstrate that MV-Adapter can generate consistent, high-quality multi-view images beyond the six views designed for specific applications. This extension further verifies the scalability and practicality of our adapter framework, showcasing its potential for arbitrary view synthesis in diverse applications. More results can be found in the supplementary materials.

### A.4 Limitations and Future Works

**Domain gap between synthetic data and natural images.** A domain gap exists between the synthetic multi-view data rendered from 3D datasets (Deitke et al., 2023) and natural images, particularly in terms of background presence and visual fidelity. The model trained with synthetic data will be affected to some extent by the specific 3D style appearance, which may affect the generalization of the model. Although the adapter design successfully leverages the priors from the pre-trained T2I model, the quality of the generated images is still influenced by the suboptimal visual quality of the training data. A potential solution involves augmenting the training data with real video datasets,

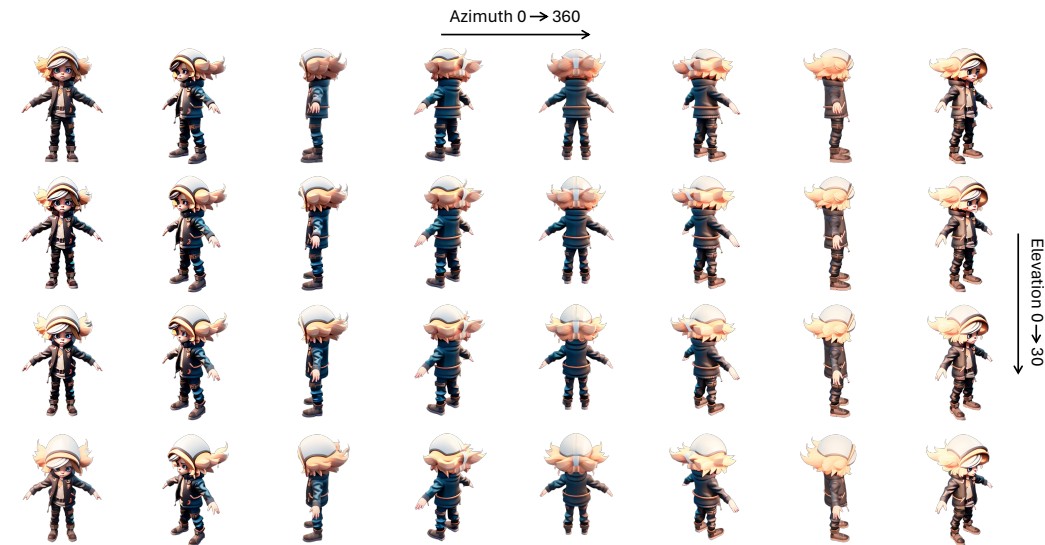

Figure 18: Visualization results using MV-Adapter to generate arbitrary viewpoints.

such as MVImgNet (Yu et al., 2023), which could reduce the domain gap. Additionally, during inference, we recommend incorporating a reference image as additional content control, which will improve the visual fidelity the controllability of the multi-view generation.

**Dependency on image backbone.** Within our decoupled attention mechanism, the visual content, multi-view consistency and alignment with the reference image originate from the underlying image backbone, multi-view attention, and image cross-attention mechanisms, respectively. Notably, both the multi-view attention and image cross-attention layers are initialized using the parameters of the original spatial self-attention layers. Consequently, the overall performance of MV-Adapter is heavily dependent on the base T2I model. If the foundational model struggles to generate content that aligns with the provided prompt or produces images of low quality, MV-Adapter is unlikely to compensate for these deficiencies. On the other hand, employing superior image backbones can enhance the synthetic results. We present a comparison of outputs generated using SDXL (Podell et al., 2024) and SD2.1 (Rombach et al., 2022) models in Fig. 19, which confirms this observation, particularly in text-conditioned multi-view generation. We believe that MV-Adapter can be further developed by utilizing advanced T2I models (Team, 2024; Labs, 2024) based on the DiT architecture (Peebles & Xie, 2023), to achieve higher visual quality in the generated images.

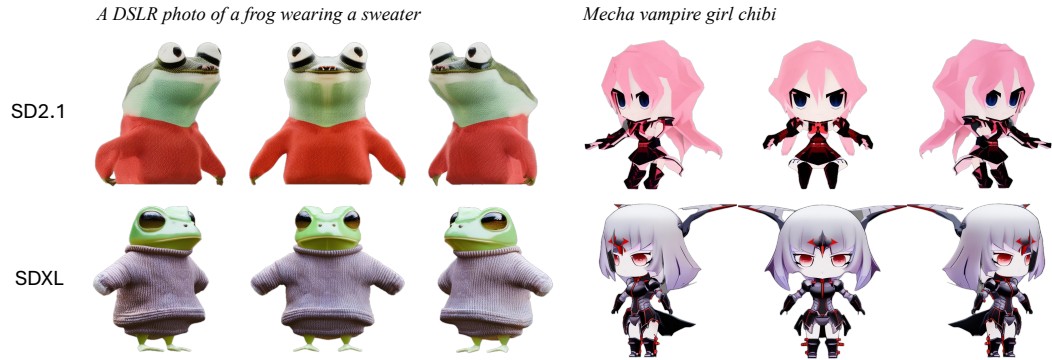

Figure 19: Qualitative comparison of our MV-Adapter based on SD2.1 and SDXL.

**Future works: sparse-view input, 3D scene generation, dynamic multi-view video generation.**
This paper provides extensive analyses and enhancements for our novel multi-view adapter, MV-

Adapter. While our model has significantly improved efficiency, adaptability, versatility, and performance compared to previous models, we identify three promising areas for future work:

- Sparse-view input. To enhance controllability in multi-view generation, we can input sparse views into our image encoder (*i.e.*, pre-trained SD U-Net), allowing multiple views to guide the generation process.

- 3D scene generation. We conducted experiments on synthetic data. Our method can be extended to scene-level multi-view generation, accommodating both camera- and geometry-guided approaches with text or image conditions.

- Dynamic multi-view video generation. Exploring dynamic multi-view video generation using a similar approach as MV-Adapter within text-to-video generation models (Zheng et al., 2024; Yang et al., 2024) presents a valuable opportunity for further advancements.

**Future works: modeling new knowledge like MV-Adapter.** By decoupling the learning of geometric knowledge from the image prior, our framework efficiently integrates new knowledge without compromising the base model's rich visual capabilities. This principle enhances learning from limited data and inspires other tasks that build upon existing image priors to learn new types of knowledge. Beyond multi-view consistency, our approach can be extended to learn zoom in/out effects, consistent lighting conditions, and other viewpoint-dependent attributes. It is possible to model viewpoint-dependent attributes such as lighting, shadows, and reflections by fine-tuning our decoupled multi-view attention on some specific small datasets, which can be defined as personalization or customization of geometric knowledge. MV-Adapter also provides insights for modeling physical or temporal knowledge based on image priors, paving the way for future research in related domains.

### A.5 MORE COMPARISON RESULTS

#### A.5.1 IMAGE-TO-MULTI-VIEW GENERATION

To provide a more in-depth analysis of our quantitative results on image-to-multi-view generation, we conducted a user study comparing MV-Adapter (based on SD2.1 (Rombach et al., 2022)) with baseline methods (Wang & Shi, 2023; Shi et al., 2023a; Wang et al., 2024b; Voleti et al., 2024; Wen et al., 2024; Li et al., 2024). The study aimed to evaluate both multi-view consistency and image quality preferences. We selected 30 samples covering a diverse range of categories, such as toy cars, medicine bottles, stationery, dolls, and sculptures. A total of 50 participants were recruited to provide their preferences between the outputs of different methods.

Participants were presented with pairs of multi-view images generated by MV-Adapter and the baseline methods. For each pair, they were asked to choose the one they preferred in terms of multi-view consistency and image quality. The results of the user study are summarized in Fig. 20. The findings indicate that, in terms of multi-view consistency, MV-Adapter performs comparably to Era3D, with preference rates of 25.07% and 22.33%, respectively. However, regarding image quality, MV-Adapter demonstrates a significant advantage, receiving a higher preference rate of 36.80% compared to the baseline methods. The improved image quality can be attributed to MV-Adapter's ability to leverage the strengths of the underlying T2I models without full fine-tuning, preserving the original feature space and benefiting from the high-quality priors of the base models.

Additionally, we provide supplementary qualitative comparison results in Fig. 21, showcasing side-by-side examples of images generated by MV-Adapter and the baseline methods. These examples further illustrate the superior image quality and consistency achieved by MV-Adapter, highlighting finer details, better texture reproduction, and more coherent structures across different views.

#### A.5.2 IMAGE-TO-3D GENERATION

To further evaluate the consistency of multi-view generation and the applicability of MV-Adapter to downstream tasks, we conducted a quantitative comparison of 3D reconstruction performance using MV-Adapter and Era3D (Li et al., 2024), which shares a similar pipeline with our method. The comparison was performed on the Google Scanned Objects (GSO) dataset, focusing on metrics such as Chamfer Distance and Volumetric IoU to assess the geometric quality of the reconstructed 3D models.

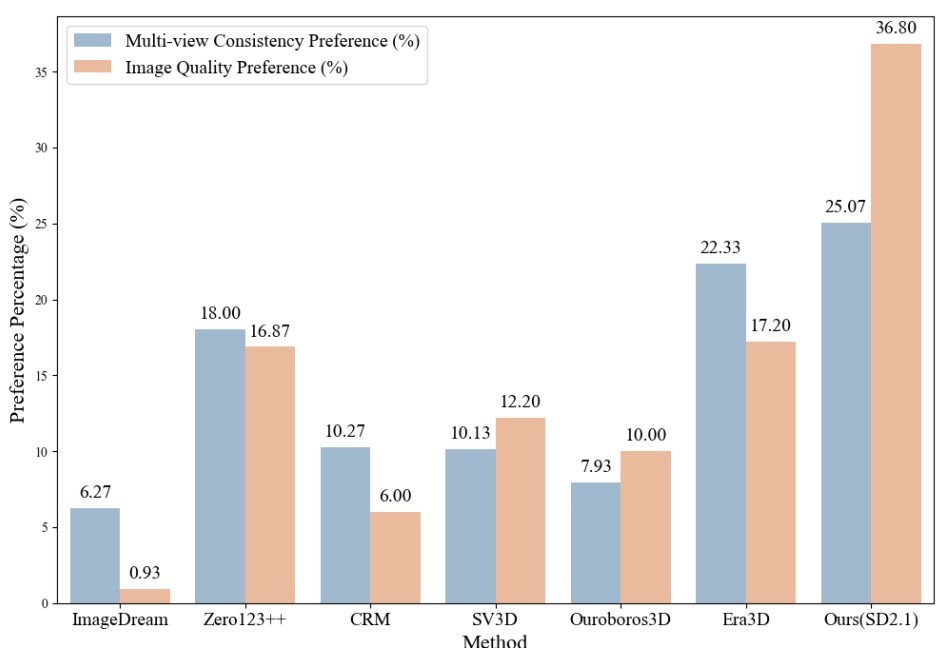

Figure 20: Results of user study on image-to-multi-view generation.

The results, summarized in Table 7, show that 3D reconstruction quality using MV-Adapter based on Stable Diffusion 2.1 (SD2.1) is comparable to that achieved with Era3D. However, when using MV-Adapter based on Stable Diffusion XL (SDXL), the reconstruction quality is significantly higher, with notable improve-

Table 7: Quantitative comparison on 3D reconstruction.

| Method | Chamfer Distance↓ | Volume IoU↑ |
|---|---|---|
| Era3D | 0.0329 | 0.5118 |
| Ours (SD2.1) | 0.0317 | 0.5173 |
| Ours (SDXL) | **0.0206** | **0.5682** |

ments in both Chamfer Distance and Volumetric IoU. This demonstrates that MV-Adapter's efficient training design facilitates compatibility with larger and more advanced base models, such as SDXL, thereby delivering superior results in 3D reconstruction tasks. These findings underline the scalability of MV-Adapter and its ability to leverage the strengths of state-of-the-art T2I models, providing additional benefits to downstream tasks like 3D generation.

## A.6 MORE VISUAL RESULTS

In Fig. 22 and Fig. 23, we show more visual results of MV-Adapter on camera-guided text-to-multiview generation with community models and extensions, such as ControlNet (Zhang et al., 2023) and IP-Adapter (Ye et al., 2023). In Fig. 24, we show more visual results on camera-guided image-to-multiview generation. In Fig. 25, we show more visual results on text-to-3D generation. In Fig. 26, we show more visual results on image-to-3D generation. In Fig. 27, we show more visual results on geometry-guided text-to-texture generation. In Fig. 28, we show more visual results on geometry-guided image-to-texture generation. Note that we have removed the background of the generated images in the visual results.

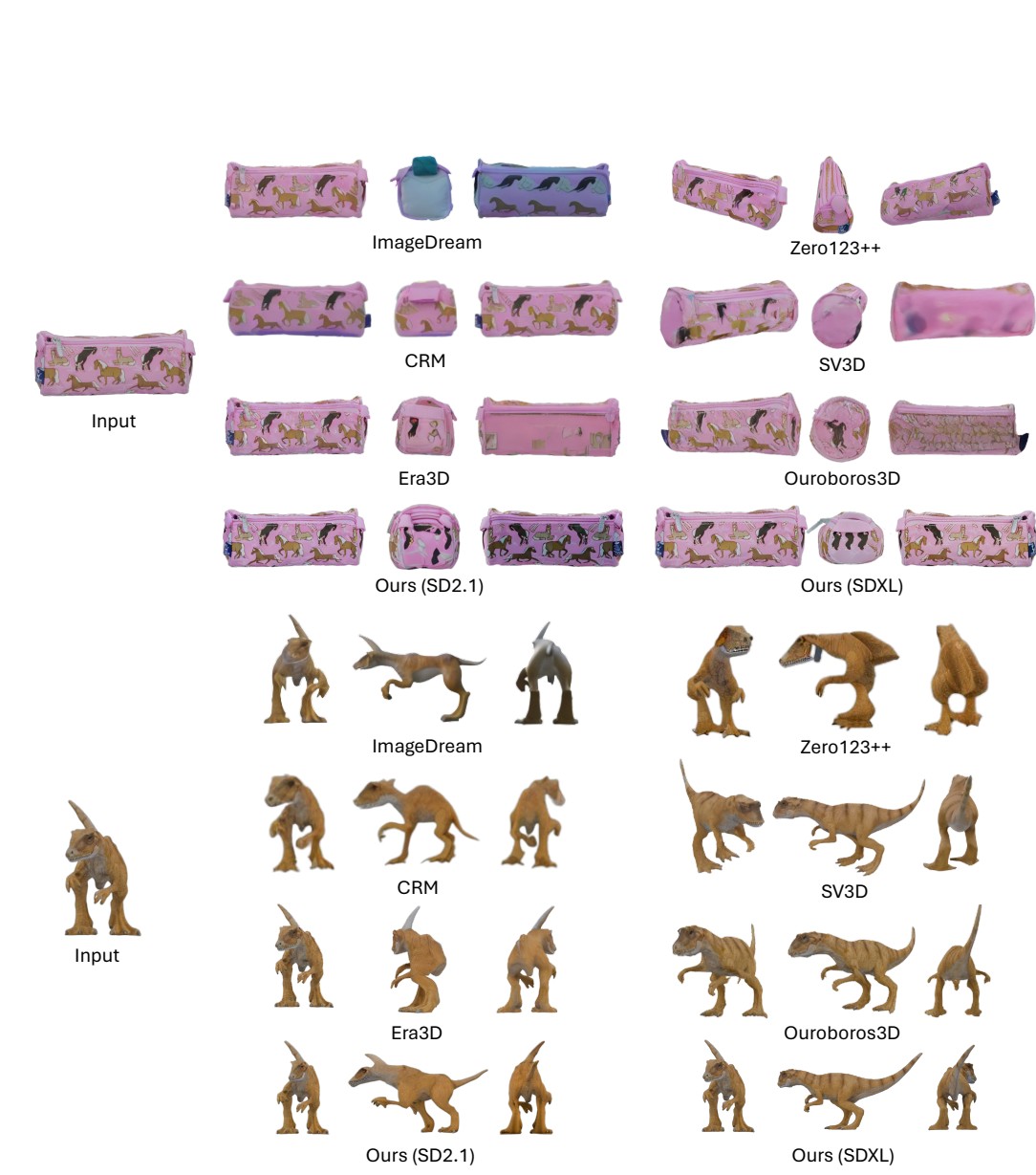

Figure 21: More qualitative comparison on image-to-multiview generation.

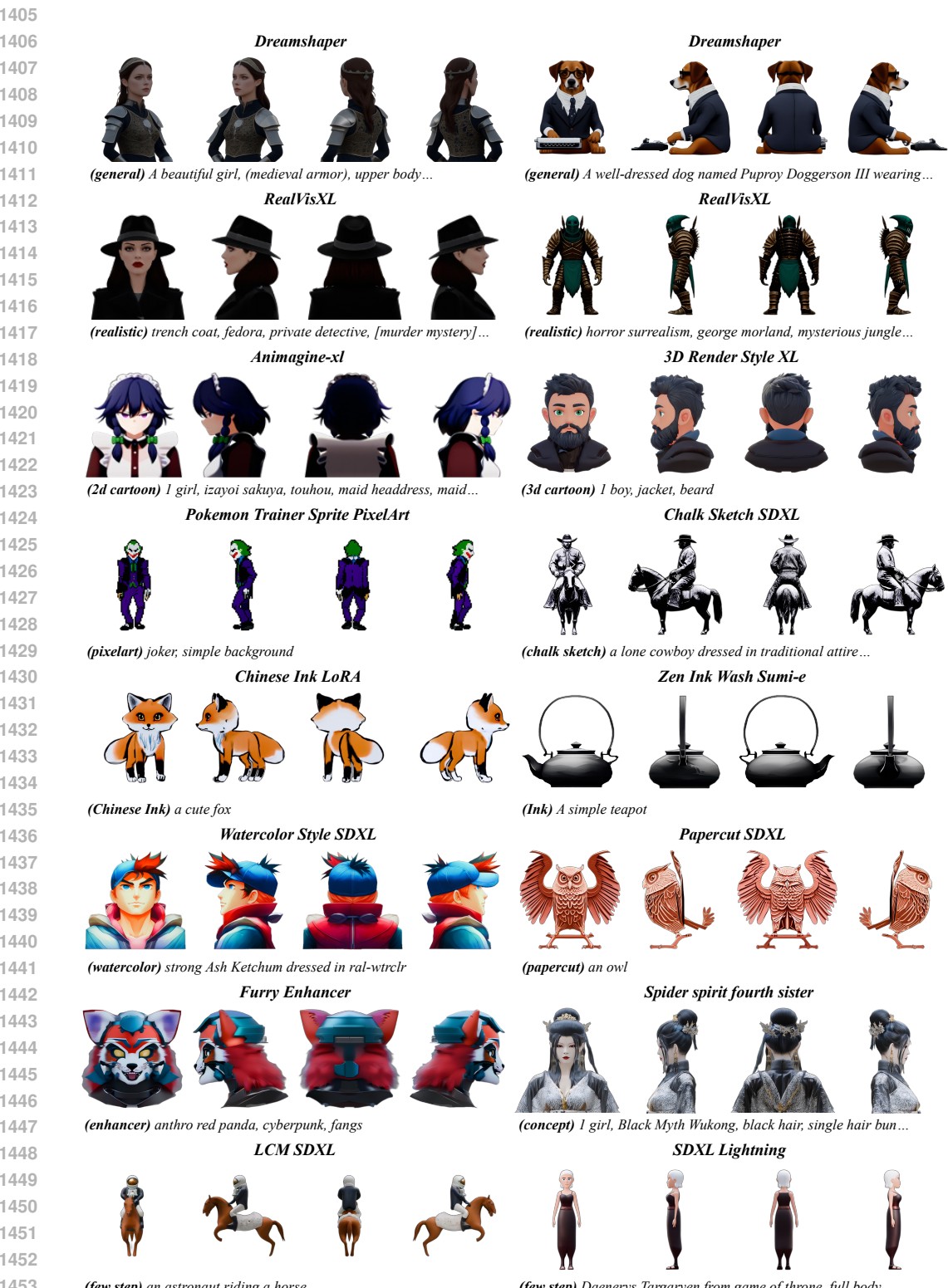

Figure 22: Additional results on camera-guided text-to-multiview generation with community models.

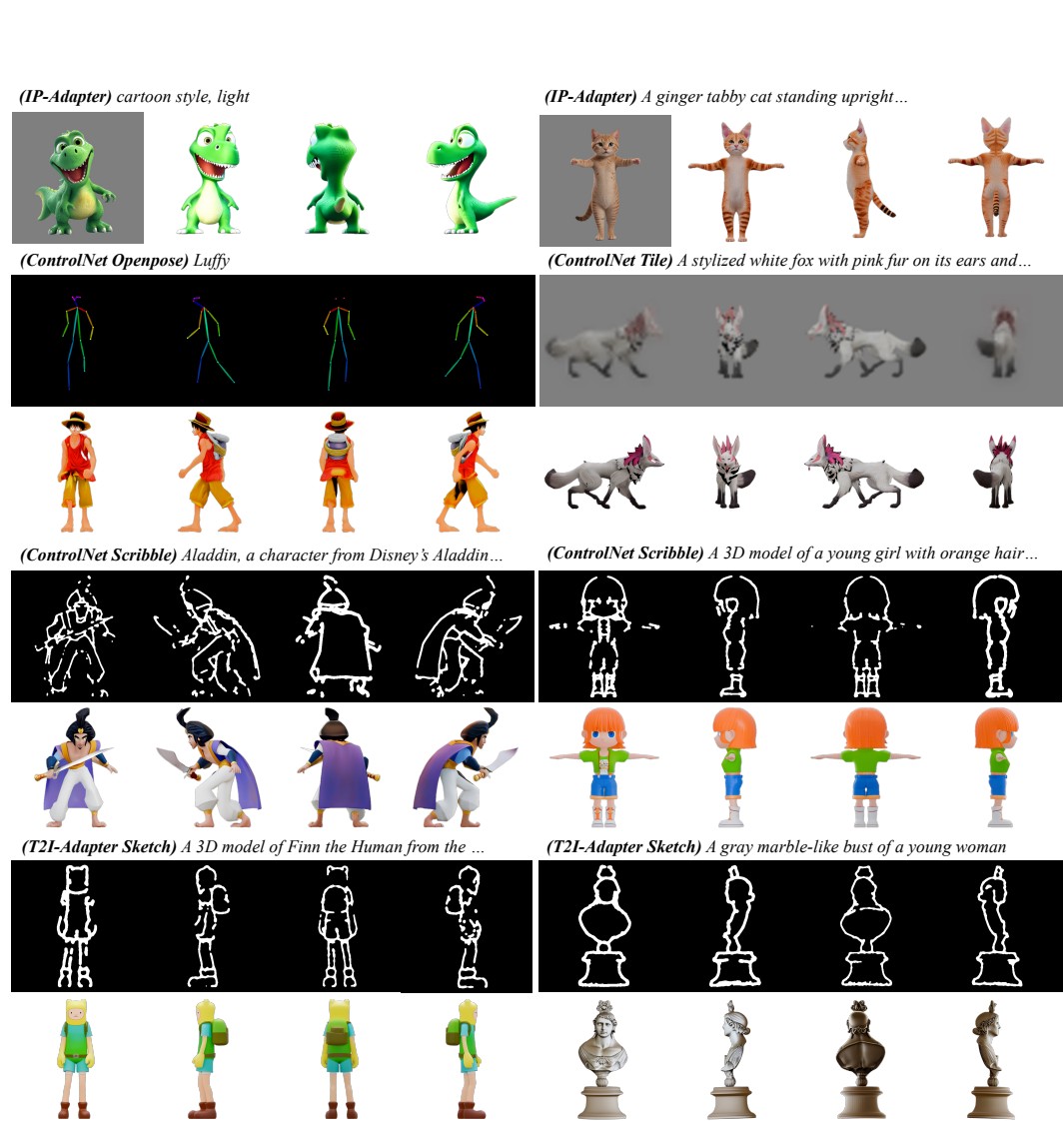

Figure 23: Additional results on camera-guided text-to-multiview generation with extensions.

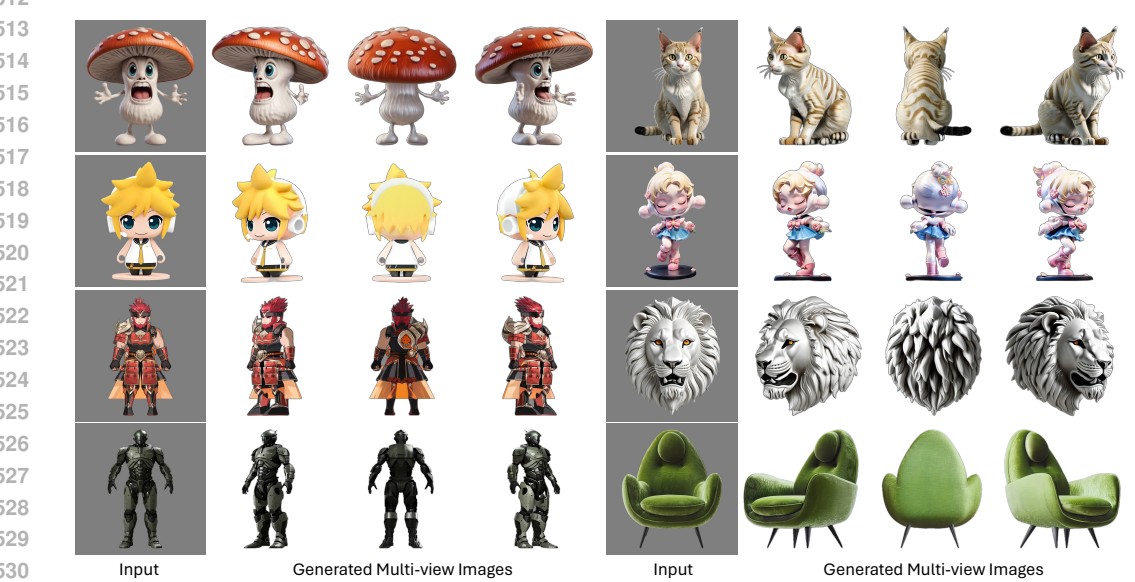

Input          Generated Multi-view Images          Input          Generated Multi-view Images

Figure 24: Additional results on camera-guided image-to-multiview generation.

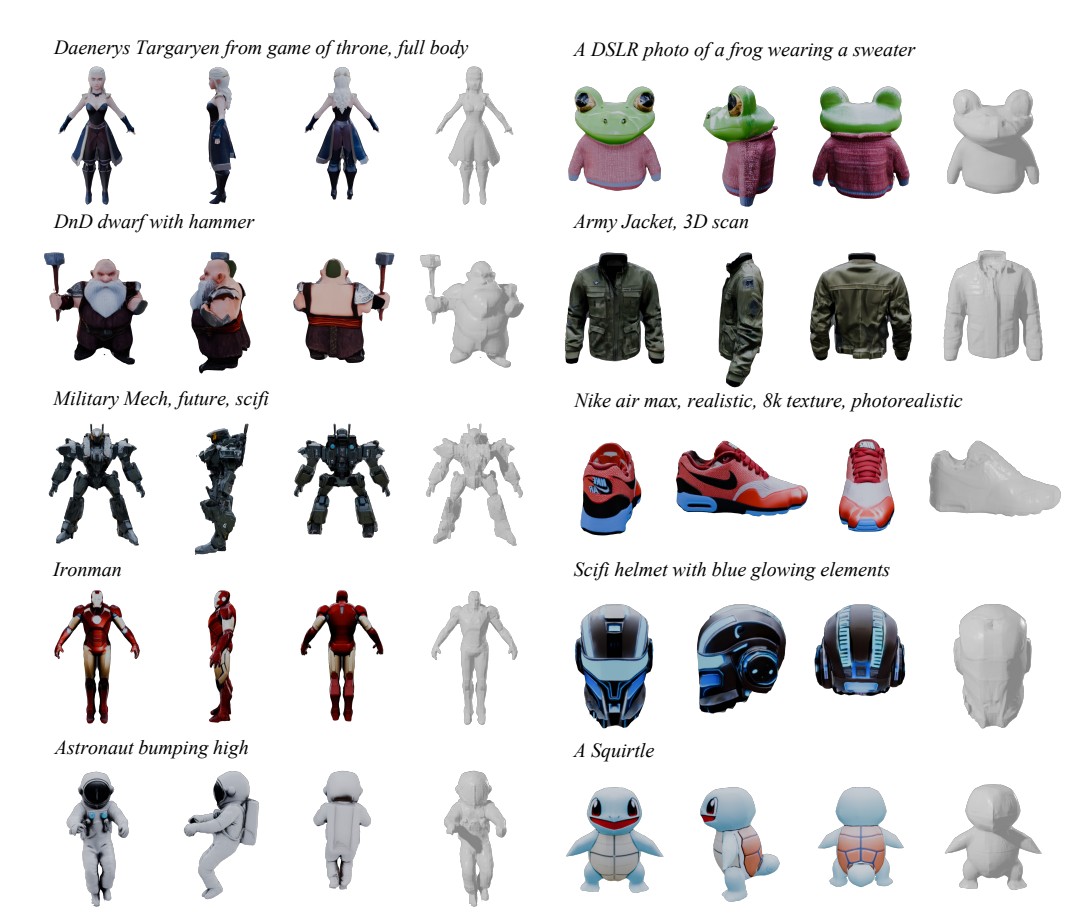

Figure 25: Visual results on text-to-3D generation.

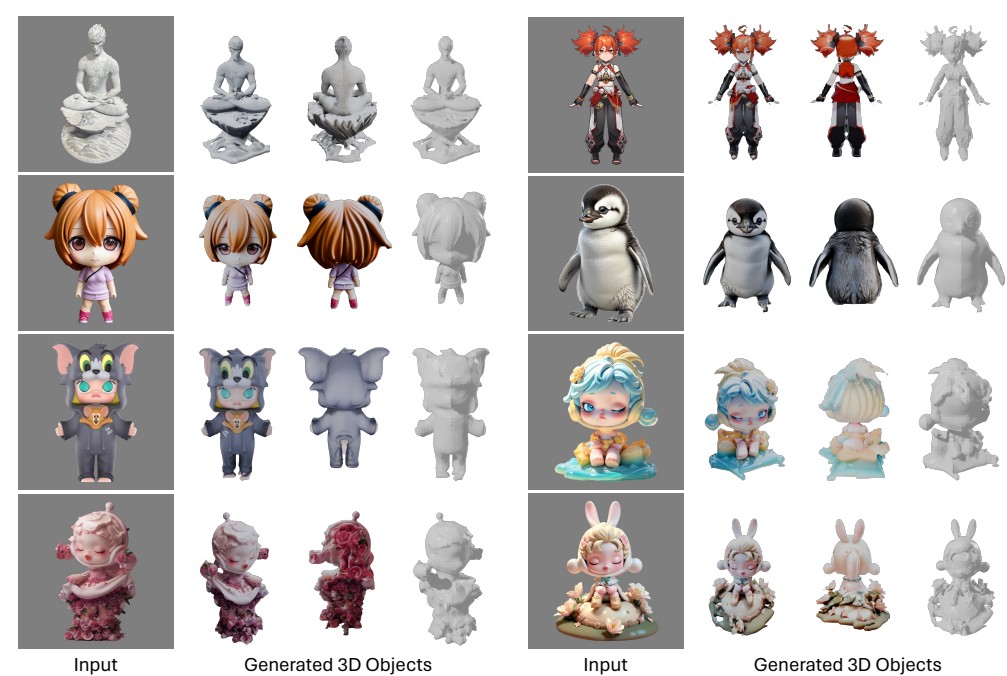

Input        Generated 3D Objects        Input        Generated 3D Objects

Figure 26: Visual results on image-to-3D generation.

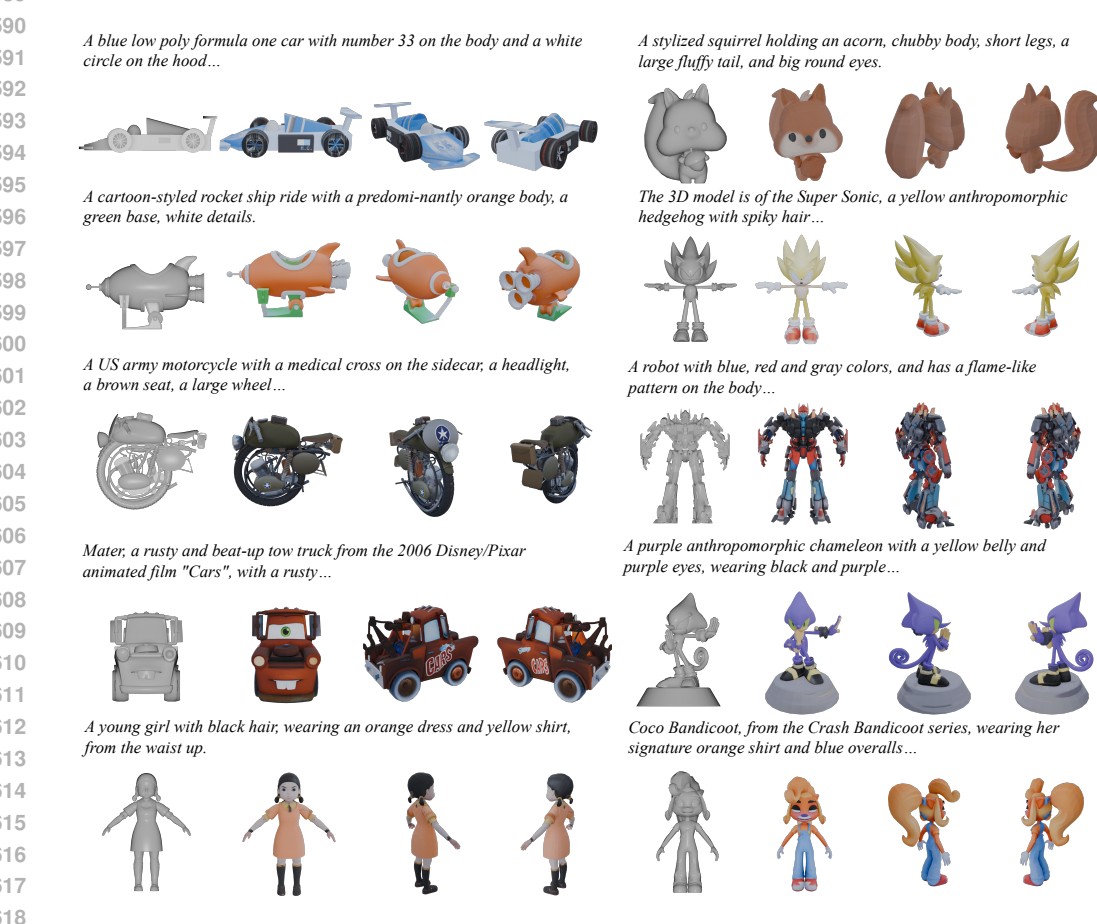

Figure 27: Additional results on geometry-guided text-to-texture generation.

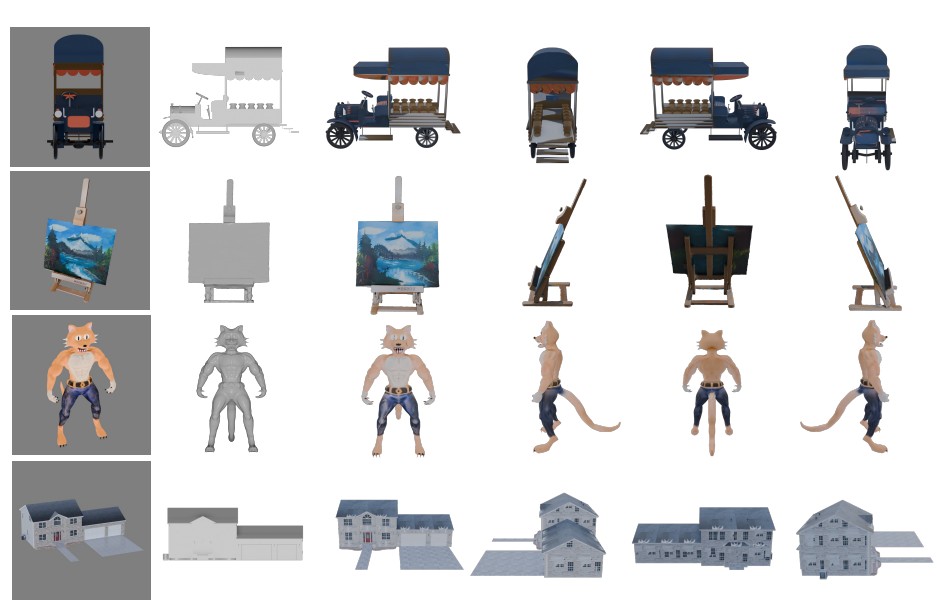

Input                    Rendered Multi-view Images from Generated Texture

Figure 28: Additional results on geometry-guided image-to-texture generation.

