# OpenReview forum: "MV-Adapter: Multi-view Consistent Image Generation Made Easy"
_ICLR.cc/2025/Conference — Submitted to ICLR 2025_

### Official Review · Reviewer_wkdR · 2024-10-18

**Soundness:** 3
**Presentation:** 3
**Contribution:** 2
**Rating:** 5
**Confidence:** 4

**Summary:**

This paper proposes a new method for consistent multi-view image generation. The proposed method followed the core idea of T2I-Adapter and made some extensions to it (e.g., conditions, training data) to be applied to the multi-view image generation task. The experimental results demonstrate the effectiveness of the proposed method.

**Strengths:**

- This paper is well-written and easy to follow.
- The results are visually pleasing and show improvement against previous methods.

**Weaknesses:**

- My major concern is that the technical contributions of this paper are a bit thin. Specifically, this paper can be viewed as an extension of T2I-Adapter [1] to the consistent multi-view image generation task. Therefore, most of the merits of the proposed method (e.g., efficiency, adaptability, versatility, performance) are inherited from the T2I-Adapter but are not original. The true contributions of this paper are actually the modifications to T2I-Adapter, which are incremental.

[1] Mou, C., Wang, X., Xie, L., Wu, Y., Zhang, J., Qi, Z. and Shan, Y., 2024, March. T2i-adapter: Learning adapters to dig out more controllable ability for text-to-image diffusion models. In Proceedings of the AAAI Conference on Artificial Intelligence (Vol. 38, No. 5, pp. 4296-4304).

- For the modifications to T2I-Adapter, the conditions (Sec. 4.1) and attentions (Sec. 4.2) are mostly borrowed from previous works (the references are included in the paper) and there are no significant new insights as well.

Therefore, I think the paper is publishable but maybe not at a top conference like ICLR.

**Questions:**

Please see the weaknesses mentioned above.

---

> ### Author Response · Authors · 2024-11-21
>
> We appreciate your review but respectfully disagree with the characterization of our contributions as merely incremental modifications to T2I-Adapter. While T2I-Adapter is a component within our framework, our core contributions extend significantly beyond it.
>
> **Our core contributions are summaried as follows:**
> 1. **First Adapter-Based Solution for Multi-View Image Generation.** We introduce the first adapter framework specifically designed for multi-view image generation—a challenging task that greatly benefits from this approach. The advantages you mentioned (efficiency, adaptability, versatility, performance) are not simply inherited from T2I-Adapter. Instead, they stem from the adapter concept and our innovative design within this framework.
> 2. **Technical Innovations in Adapter Design.** We propose a novel parallel attention architecture that efficiently decouples the learning of geometric knowledge from image priors (**our ablation studies on L475~485 and Appendix A.3.4** thoroughly discuss and validate its effectiveness), and a unified condition embedding and encoder that enhances the model's flexibility.
> 3. **Extensions to Arbitrary View Generation and Inspirations for Future Work.** MV-Adapter can easily extend to **arbitrary view generation**,  facilitating a wider range of downstream tasks. Details are provided in **Appendix A3.6**. Furthermore, MV-Adapter **provides insights** for modeling new knowledge (such as geometric, physical or temporal knowledge) based on image priors with our framework for decoupled learning.
>
> We kindly refer you to **our global response** for a deep understanding of our work. We firmly believe that these contributions demonstrate significant innovation and provide valuable insights to the field. Our work offers a novel and effective approach to multi-view image generation that goes beyond incremental changes to existing methods. We hope this clarifies our position and the substantial contributions of our work. Thank you for your consideration.

---

> ### Author Response · Authors · 2024-11-25
>
> Dear Reviewer:
>
> Once again, we sincerely appreciate the time and effort you have dedicated to reviewing our paper. We kindly suggest that you visit our anonymous project page https://mv-adapter.github.io/ for a deeper understanding of our work.
>
> As the discussion period concludes in two days, we would be grateful if you could review our rebuttal at your convenience, should your schedule allow. If there are any further points requiring clarification or improvement, please be assured that we are fully committed to addressing them promptly. Thank you once again for your invaluable feedback on our research.
>
> **Meeting ICLR Standards**
>
> We would also like to take this opportunity to address the comment in your initial review: “Therefore, I think the paper is publishable but maybe not at a top conference like ICLR.” We believe our work aligns with ICLR's core emphasis on bringing "new, relevant, impactful knowledge" to the community through:
>
> - **Novel Technical Contribution:**
>   1. **First Systematic Adapter Framework for Multi-View Generation:** We introduce the first comprehensive adapter-based framework specifically designed for multi-view image generation, effectively addressing the complexities of handling multiple views and ensuring consistency.
>   2. **Innovative Design:** Our attention architecture and unified condition embedding enable efficient integration of geometric knowledge without altering pre-trained models.
>   3. **Thorough Ablation Studies:** We provide extensive ablations validating our design choices, demonstrating their impact on performance.
> - **Well-motivated Problem & State-of-the-Art Performance:**
>   1. **Addresses critical limitations:** Our work tackles significant limitations in existing methods, such as adaptability, scalability and resolution constraints, enabling high-resolution (768×768) multi-view generation on large models like SDXL.
>   2. **SOTA Results:** We achieve leading performance in multi-view generation, with improved consistency and quality.
> - **Impact and Utility for the Community:**
>   1. **Adaptability and Versatility:** Our MV-Adapter seamlessly integrates with various powerful or customized T2I model variants, enabling consistent multi-view generation and supporting multiple conditional inputs, which facilitates downstream applications like 3D generation and 3D texture synthesis.
>   2. **Inspiring Future Research:** Our approach of learning new knowledge based on existing priors can inspire the modeling of other geometric (e.g., lighting effects), physical, and temporal knowledge, potentially benefiting a wide range of applications.
>
> We hope that these points clarify how our work meets the standards of ICLR and contributes valuable advancements to the field. We respectfully request reconsideration of your assessment of our paper. If you have any further concerns or questions, we would be grateful for the opportunity to address them.
>
> Warm regards,
> The Authors

---

> > ### Comment · Reviewer_wkdR · 2024-11-25
> > **Thank you for your rebuttal but my concerns about novelty remain**
> >
> > Dear authors,
> >
> > Thank you for your rebuttal but my concerns about novelty remain.
> >
> > As mentioned in my review, my concern is about the *technical* contribution of the work. Combining multiple existing techniques to address a new application is okay but this should come with new insights, which I did not see much in this paper. The rebuttal does not specify *technically* which part is novel and how they are novel.
> >
> > Therefore, I would keep my original rating.

---

> > > ### Author Response · Authors · 2024-11-30
> > >
> > > Dear Reviewer,
> > >
> > > Thank you for your time and effort in reviewing our work. We sincerely appreciate your detailed feedback, which has helped us refine our responses. As the discussion period concludes in a few days, we would be grateful if you could review our second-round rebuttal at your convenience.
> > >
> > > Thank you again for your thoughtful consideration.
> > >
> > > Warm regards, The Authors

---

> ### Author Response · Authors · 2024-11-25
>
> We appreciate your focus on technical novelty. Let us precisely articulate the technical contributions in our work:
>
> **Technical Contributions**
>
> Our main contribution lies in the **technically innovative adapter framework** we propose for multi-view image generation, **rather than in specific components like T2I-Adapter or Row-wise Attention**.
>
> 1. **Attention Architecture.** We propose novel attention architecture to preserve the integrity of the base T2I model while enabling efficient learning of geometric knowledge through a parallel structure.
>     - **Technical details:**
>       - **Duplication of Self-Attention Layers:** Our design adheres to the principle of preserving the original network structure and feature space of the base T2I model. Existing methods like MVDream and Zero123++ modify the base model's self-attention layers to include multi-view or reference features, which disrupts the learned priors and requires full model fine-tuning. In contrast, we duplicate the self-attention layers (including architecture and weights) to create new multi-view attention and image cross-attention layers, initializing the output projections to zero. This allows the new layers to learn geometric knowledge without interfering with the original model's priors, ensuring excellent adaptability.
>       - **Parallel Attention Structure:** The straightforward way to insert new layers is to append them serially after the original layers. However, it requires the new layers to learn from scratch and diminishes the effectiveness of pre-trained priors. Even if we initialize the new layers with the pre-trained self-attention parameters, the features input to new layers are in a different domain with those input to original self-attention layers, causing the initialization to be ineffective. Therefore, we design a parallel architecture. This ensures that the new layers can fully inherit the powerful priors of the pre-trained self-attention layers, enabling efficient learning of geometric knowledge.
>     - **Empirical Validation:** Our ablation studies and experimental results validate the effectiveness of our design:
>       - **Figures 1, 5, 8, and 22** in our paper display visual results that demonstrate our adapter's strong adaptability and effectiveness. And **Figure 9** provides visual comparisons with existing methods on the adaptability, showcasing the advantages of our adapter design.
>       - **Table 5 and Figure 17** presents quantitative and qualitative results of ablation study on attention architecture, demonstrating the superiority of our parallel design.
> 2. **Unified Condition Embedding and Encoder.** We introduce a unified condition embedding and encoder that seamlessly integrates camera parameters and geometric information into spatial map representations, enhancing the model's flexibility and applicability.
>     - **Technical details:** To enable the adapter's versatility—supporting both camera conditions and geometry conditions for critical 3D applications like 3D generation and 3D texture generation—we innovatively unify camera parameters and geometric information into spatial map representations. We introduce a spatial-aligned condition encoder that encodes both types of maps using the same architecture. Previous methods have only encoded either camera information or geometric information but have not unified both within a single framework. By adopting this unified representation and encoder, our adapter gains enhanced flexibility and applicability, seamlessly handling various types of guidance within a unified framework.
>     - **Empirical Demonstration:** Figures 6 and 7 in the paper illustrate camera-guided multi-view image generation, while Figures 8 and 10 demonstrate geometry-guided multi-view image generation and texture generation. These results confirm that our design enables the model to support multiple functionalities, which previous works have not achieved.
>
> **Valuable Insights**
>
> - **Decoupling Image Prior and Geometric Knowledge:** By decoupling the learning of geometric knowledge from the image prior, our framework efficiently integrates new knowledge without compromising the base model's rich visual capabilities.
> - **Inspiration for Future Research:** This principle not only enhances learning from limited data but also inspires other tasks that build upon existing image priors to learn new types of knowledge. Our approach can be applied to learning **zoom in/out effects, consistent lighting, and other viewpoint-dependent properties**. It provides valuable insights for **modeling physical or temporal knowledge** based on image priors.
>
> Each of these represents a concrete technical advancement, not just a combination of existing techniques. We respectfully request that you reconsider your assessment of our paper.
>
> Thank you once again for your thoughtful feedback. If you have any further questions or concerns, we would be happy to address them.

---

### Official Review · Reviewer_VRDn · 2024-10-27

**Soundness:** 2
**Presentation:** 3
**Contribution:** 2
**Rating:** 6
**Confidence:** 4

**Summary:**

This paper introduces MV-Adapter, a plug-and-play adapter designed to support various conditions when generating multi-view consistent images. It preserves the base T2I model’s priors, improving efficiency by eliminating the need for constant full fine-tuning. Additionally, MV-Adapter supports multiple conditioning inputs, enhancing versatility. All the contributions above make the performance better.

**Strengths:**

1. MV-Adapter supports various conditions, which provides a possible solution for multi-view generation with more conditional control.
2. MV-Adapter is more efficient and has less training costs according to the data shown in the paper.

**Weaknesses:**

1. The paper does not present the metrics results of 3D reconstruction using the generated multiviews, a downstream task that most directly reflects the consistency of the generated views. It would be beneficial to include evaluations using standard 3D reconstruction metrics such as volumetric IoU or Chamfer distance. This would provide quantitative insights into the multi-view consistency and help validate the quality of the generated outputs.
2. The proposed “image cross-attention” mechanism is not novel, as it has already been employed in Zero123++. It would be helpful to explicitly compare the proposed method with the implementation in Zero123++, focusing on any specific differences or improvements.

**Questions:**

1. How does the result of reconstructing 3D with MV-Adapter method compare with other multi-view generation methods? Does it offer any distinct advantages, or are there potential limitations that need to be addressed?
2. I noticed that the paper does not mention whether the text captions used for training are sourced directly from the Cap3D dataset or if they were separately annotated by the authors. Could you clarify the origin of these text captions.
3. For multi-view generation, the primary focus should be on the quality of the generated outputs rather than merely reducing the number of trainable parameters. I am curious how the model would perform if subjected to full-model fine-tuning while keeping the current design unchanged. This could further validate the architecture by demonstrating that comparable results can be achieved without the need to fine-tune all parameters, providing additional insights into the effectiveness of the proposed design.

---

> ### Author Response · Authors · 2024-11-21
> **Response (1)**
>
> Thank you for your valuable feedback and for highlighting areas where we can improve our work. We address your concerns and questions point by point below.
>
> **1. Quantitative Results of 3D Reconstruction for Validating Multi-view Consistency.**
>
> To further evaluate the consistency of our multi-view generation and its applicability to downstream tasks, we have added a quantitative comparison of 3D reconstruction on the GSO dataset using MV-Adapter and Era3D (which shares a similar pipeline). As shown in **Table 7 (see Appendix A5.2)**, the 3D reconstruction quality using MV-Adapter based on SD2.1 is comparable to that achieved with Era3D. Importantly, when using MV-Adapter based on SDXL, the reconstruction quality improves significantly, with notable enhancements in both Chamfer Distance and Volumetric IoU metrics. These results demonstrate the scalability of MV-Adapter and its ability to leverage the strengths of state-of-the-art T2I models, providing additional benefits to downstream tasks like 3D reconstruction.
> > **Table 7**
> | Method | Chamfer Distance$\downarrow$ | Volumetric IoU$\uparrow$ |
> | - | :-: | :-: |
> | Era3D | 0.0329 | 0.5118 |
> | **Ours (SD2.1)** | **0.0317** | **0.5173** |
> | **Ours (SDXL)** | **0.0206** | **0.5682** |
>
> Furthermore, to provide a more in-depth analysis of our quantitative results on image-to-multi-view generation, we conducted a user study. Details of this study and its findings are presented in **Appendix A5.1**.

---

> ### Author Response · Authors · 2024-11-21
> **Response (2)**
>
> **2. Comparison with Reference Attention in Zero123++.**
>
> While both our image cross-attention mechanism and the reference attention in Zero123++ employ a pre-trained U-Net to encode the reference image, there is a fundamental difference in how they integrate these features:
> - **Zero123++:** Extends the denoising U-Net's self-attention keys and values to include reference features, requiring full fine-tuning of the base Stable Diffusion model. This approach modifies the original network's feature space, potentially disrupting the base model's prior knowledge.
> - **Our Method:** Duplicates the self-attention layers to create new image cross-attention layers (including network structure and weights), organizing them in a parallel architecture. By initializing the output projections to zero, we decouple the learning of reference image information from the base model. This design avoids full-model fine-tuning and preserves the base model's capabilities, thereby enabling our model compatible with the **adapter principle**.
>
> Our approach integrates reference information without interfering with the pre-trained model, maintaining the integrity of the original feature space while efficiently learning from the reference images.
>
> **3. Advantages of Using MV-Adapter to reconstruct 3D.**
>
> The advantages of using MV-Adapter for 3D reconstruction lie in its adaptability and scalability:
>
> - **Adaptability:** Users can attach MV-Adapter to various base model variants or customized models (e.g., with DreamShaper, stylized or personalized models, ControlNet) to achieve higher-quality, customized, and controllable multi-view generation, as demonstrated in our paper.
> - **Scalability:** The efficient training of MV-Adapter makes it feasible to scale to larger and more powerful T2I base models like SDXL. This scalability enables modeling stronger consistency and generating higher-quality multi-view images, providing significant benefits to downstream tasks like 3D reconstruction.
>
> These advantages enhance the usability and effectiveness of MV-Adapter in practical applications requiring high-quality 3D reconstructions from multi-view images.
>
> **4. Text Captions Used for Training.**
>
> The text captions used for training are sourced directly from the Cap3D dataset. We have clarified this point by adding a statement in the **"Dataset" section of Appendix A2 (Implementation Details)**.
>
> **5. Consideration of Full-Model Fine-Tuning.**
>
> We believe that efficient training (i.e., reducing the number of trainable parameters) is also important in multi-view generation tasks for several reasons:
> - **Potential for Stronger Base Models:** Efficient training allows adaptation to more powerful base T2I models. As shown in our results, switching the base model to SDXL brings significant gains in multi-view consistency and image quality.
> - **Preservation of Base Model's Feature Space:** By avoiding modifications to the base model's original feature space, MV-Adapter exhibits strong adaptability, compatible with various model variants and plugins. This compatibility enhances controllability and usability in multi-view generation.
>
> To address your suggestion, we conducted an experiment on text-to-multiview generation, comparing our MV-Adapter (SD2.1) with a version using full-model fine-tuning while keeping the current design unchanged. As shown in **the following table**, full-model fine-tuning leads to overfitting on the 3D dataset, resulting in poorer performance. This demonstrates that our approach, which avoids full-model fine-tuning, maintains the base model's generalization capabilities and achieves superior results without the need to fine-tune all parameters.
>
> > **Table R1**
> | Setting | FID$\downarrow$ | IS$\uparrow$ | CLIP Score$\uparrow$ |
> | - | :-: | :-: | :-: |
> | MV-Adapter (SD2.1, Full-tuning) | 33.76 | 13.23 | 30.51 |
> | **MV-Adapter (SD2.1, Ours)** | **31.24** | **15.01** | **32.04** |
>
> We hope these clarifications and additional results address your concerns and provide a deeper understanding of our work. Thank you again for your constructive feedback.

---

> > ### Comment · Reviewer_VRDn · 2024-11-24
> >
> > Thanks to the authors for the response. I appreciate the additional comparison experiments between MV-Adapter (SD2.1, Full-tuning) and MV-Adapter (SD2.1, Ours). However, I noticed that the provided metrics do not include any 3D-consistency-related evaluations. I believe it would be valuable to supplement the experiments with such metrics.
> >
> > It is possible that Full-tuning could exhibit stronger 3D consistency, which might be expected. I am curious about the trade-off in this context and would like to understand how your approach balances this aspect.

---

> > > ### Author Response · Authors · 2024-11-25
> > >
> > > Thank you for your valuable feedback. In our initial rebuttal, we provided comparison experiments under the text condition setting, primarily evaluating the **quality** of multi-view generation. Responding to your suggestion, we have now supplemented our experiments with image condition-based multi-view generation evaluations on the GSO dataset.
> > >
> > > We computed relevant 3D-consistency-related metrics, including both image-based metrics and 3D reconstruction metrics. As shown in the following table, our MV-Adapter, even with only adapter fine-tuning, achieves multi-view consistency **comparable** to the fully fine-tuned version. These results are attributable to our proposed **attention architecture**, where we duplicate the self-attention layers (including architecture and weights) to create new layers and organize them in a parallel way. As demonstrated by **Table 5 and Figure 17 in the paper**, this approach allows the **adapter** to **inherit powerful image priors while efficiently learning geometric knowledge**.
> > >
> > > > **Table R2**
> > > | Setting | PSNR$\uparrow$ | SSIM$\uparrow$ | LPIPS$\downarrow$ | Chamfer Distance$\downarrow$ | Volumetric IoU$\uparrow$ |
> > > | - | :-: | :-: | :-: | :-: | :-: |
> > > | MV-Adapter (SD2.1, Full-tuning) | **20.934** | 0.8602 | 0.1187 | **0.0306** | 0.5142 |
> > > | **MV-Adapter (SD2.1, Ours)** | 20.867 | **0.8695** | **0.1147** | 0.0317 | **0.5173** |
> > >
> > > These findings, including our image-quality-related evaluations in the initial rebuttal **(Table R1)** and supplemented 3D-consistency-related evaluations **(Table R2)**, as well as our ablation study on our architectural design **(Table 5 in the paper)**, reinforce the effectiveness of our technical innovations on **attention architecture**, which contributes to the robustness of our MV-Adapter in balancing 3D consistency with performance.
> > >
> > > Thank you once again for your constructive feedback. If you have any further questions or require additional clarifications, we are more than willing to address them.

---

> > > > ### Comment · Reviewer_VRDn · 2024-11-26
> > > >
> > > > Thank you for providing the additional experimental results. The new data and analysis have addressed my concerns effectively. I am now satisfied with the clarification and will be adjusting my score accordingly to reflect the improvement.

---

### Official Review · Reviewer_C83o · 2024-10-28

**Soundness:** 4
**Presentation:** 3
**Contribution:** 3
**Rating:** 6
**Confidence:** 4

**Summary:**

This paper introduces MV-adapter, a plug-and-play module for pretrained T2I models, enabling them to support multi-view generation. Beyond the base image generation models, the proposed adapter can be integrated with other custom models, such as ControlNet. It allows for camera pose guidance, geometry guidance, and can optionally use an image as a condition. The authors conduct extensive experiments on various use cases and perform ablation studies to investigate network design choices.

**Strengths:**

1. The paper is well-written, clearly organized, and easy to follow.
2. The experiments are comprehensive. The proposed method is tested on multiple base networks, including SD2.1, SDXL, and other fine-tuned models derived from these two.
3. The proposed method supports both geometry-guided and camera pose-guided generation, significantly expanding its potential real-world applications.
4. The module can be integrated with ControlNet, reusing the control signals learned from ControlNet.

**Weaknesses:**

While I do not observe any major weaknesses in the paper, I still have some questions regarding the implementation and experimental details:

1. The paper does not provide much detail about the T2I-UNet used for image cross-attention. Is this UNet identical to the diffusion UNet? What timestep is used for feature extraction?

2. The proposed method is very similar to T2I-adapter in terms of network structure. Can the MV-adapter be combined with a learned T2I-adapter, similar to ControlNet?

3. How is the CFG calculated at inference time, since the network now has multiple conditions.

**Questions:**

Please refer to the weaknesses listed above.

---

> ### Author Response · Authors · 2024-11-21
>
> Thank you for your positive feedback regarding the clarity of our paper, the comprehensiveness of our experiments, and the versatility of our proposed method.
>
> **1. Details about T2I-UNet for Image Cross-Attention.**
>
> Yes, the T2I U-Net used for extracting reference image features is identical to the diffusion U-Net. We input the reference image without noise and set the timestep to 0 to extract the features. We have added these details to the **methodology section of our revised paper**.
>
> **2. Similarity to T2I-Adapter and Possibility of Combination.**
>
> While our condition guider component resembles that of the T2I-Adapter, our overall adapter framework is innovative. We kindly refer you to **our global response** for a deeper understanding of our contributions. Additionally, our MV-Adapter can indeed be combined with a pre-trained T2I-Adapter. We have included results demonstrating this combination in **Figure 22 of the Appendix**.
>
> **3. Calculation of CFG with Multiple Conditions.**
>
> Our method for calculating Classifier-Free Guidance (CFG) with multiple conditions is similar to that used in TOSS [1]. Let `pred_uncond`, `pred_image`, and `pred_image_text` denote the model's predictions under unconditional input, image-only condition, and both image and text conditions, respectively. The CFG calculation is as follows:
> ```
> pred_uncond + cfg_image * (pred_image - pred_uncond) + cfg_text * (pred_image_text - pred_image)
> ```
> We have provided a formal description of this calculation in the **"Inference" paragraph of Appendix A.2**.
>
> [1] TOSS: High-quality Text-Guided Novel View Synthesis from a Single Image [ICLR 2024]

---

> > ### Comment · Reviewer_C83o · 2024-11-22
> > **Thank you for the response.**
> >
> > My concerns have been well addressed in the response. Therefore, I will maintain my original rating of acceptance.

---

> > > ### Author Response · Authors · 2024-11-22
> > >
> > > We sincerely thank you for your thorough review and thoughtful feedback on our paper.
> > >
> > > We are greatly encouraged by your positive assessment, particularly:
> > > - The "excellent" rating for technical soundness
> > > - The paper being "well-written" and "clearly organized"
> > > - The "comprehensive" experiments
> > > - "No major weaknesses" in the work
> > > - Our responses having "well addressed" your concerns
> > >
> > > Given these positive remarks and your final assessment of "acceptance", we respectfully suggest considering a higher score that better reflects your encouraging evaluation.
> > >
> > > We appreciate your time and consideration.

---

### Official Review · Reviewer_v3NF · 2024-10-29

**Soundness:** 3
**Presentation:** 3
**Contribution:** 2
**Rating:** 6
**Confidence:** 4

**Summary:**

This paper introduces MV-Adapter, a plug-and-play module for text-to-image diffusion models that facilitates efficient 6-view image generation with multiple conditions. MV-Adapter overcomes the limitations of existing methods, such as high computational costs, incompatibility with T2I derivatives, and limited versatility in handling various conditioning signals. It achieves this through a condition guider that processes camera pose or geometry information and a decoupled attention mechanism that incorporates multi-view and image cross-attention. The authors evaluate MV-Adapter on various T2I models and derivatives, demonstrating its effectiveness in generating high-quality multi-view images from text, image, or geometry prompts.

**Strengths:**

1: The paper is well-written and easy to follow.

2: The motivation is straightforward and intuitive. The study explores various design choices for text-to-multi-view diffusion models, making progress in all examined aspects.

3: The experiments are solid. Qualitative results showcase promising visual fidelity and view consistency across various T2I models and tasks. Quantitative evaluation also demonstrates competitive performance compared to baselines.

**Weaknesses:**

1: A key limitation is the fixed 6-view output of MV-Adapter. While sufficient for some tasks, it falls short compared to video-diffusion models like Emu-Video used in im-3d and vfusion3d (16 views) or SV3D (20 views). Additionally, MV-Adapter cannot synthesize novel views from arbitrary angles, a capability demonstrated by SV3D and Cat3D. This restricts its use in applications requiring more comprehensive 3D understanding or flexible viewpoint control.

2: The quantitative improvements over baselines appear modest (with GSO image-to-multi-view). More compelling evidence is needed to showcase the impact of MV-Adapter. A user study or an anonymous webpage hosting video comparisons for direct assessment would strengthen these claims.

3: Minor: The proposed components draw heavily from existing work, which may lead to a perceived lack of novelty. The condition guider is inspired by T2I-Adapter, and the decoupled attention borrows from Era3D.

**Questions:**

1: The paper primarily focuses on generating a fixed set of views. Did the authors considered enabling novel view synthesis with arbitrary viewing angles, similar to methods like SV3D or Cat3D? If so, how to envision adapting MV-Adapter to achieve this?

2:  Could the authors please provide more details about the experimental setup for the image-to-multiview comparison on the GSO dataset? Specifically, how many gso assets are used? Did top/bot views covered for evaluation?

I am currently leaning towards a weak rejection (more like a 4 score), and my final recommendation hinges on the authors' response to the raised weaknesses and questions in their rebuttal. Specifically, I would appreciate a discussion on the possibility of extending MV-Adapter to handle arbitrary viewpoints and a more in-depth analysis of the quantitative results.

After the rebuttal, I would like to recommend a borderline acceptance for this submission.

---

> ### Author Response · Authors · 2024-11-21
>
> Thank you for your thoughtful review and valuable feedback. We address your concerns point by point below.
>
> **1. Enabling Arbitrary View Synthesis.**
>
> While our main text focuses on generating a fixed set of six views for 3D generation and 3D texture generation, our adapter framework can be extended to synthesize multiple views from arbitrary angles. In **Appendix A3.6**, we have detailed the scheme and experiments demonstrating this capability.
>
> Our approach is similar to CAT3D: we first generate eight anchor views and then use these as conditions to guide the generation of additional views at arbitrary angles. The MV-Adapter framework remains largely unchanged—we adjust the number of input images to one or four (concatenate multiple images into a long image if the number is set to four), and replace the row-wise attention with full self-attention in the multi-view attention module. More details and experimental results are provided in **Appendix A3.6** and on our **anonymous project page** https://mv-adapter.github.io/ .
>
> **2. Stronger Quantitative Evidence on Image-to-Multi-view.**
>
> To provide a more in-depth analysis of our quantitative results, we have supplemented **Table 2** with metrics from our MV-Adapter model based on SDXL, showing significant performance improvements. Additionally, we conducted an ablation study comparing MV-Adapter (based on SD2.1) with baseline methods. We evaluated multi-view consistency and image quality preferences over 30 samples, collecting responses from 50 participants. The results and additional qualitative comparisons are documented in **Appendix A5.1**.
> > **Results of User Study**
> | Method | Multi-view Consistency Preference (%) $\uparrow$ | Image Quality Preference (%) $\uparrow$ |
> | - | :-: | :-: |
> | ImageDream | 6.27% | 0.93% |
> | Zero123++ | 18.00% | 16.87% |
> | CRM | 10.27% | 6.00% |
> | SV3D | 10.13% | 12.20% |
> | Ouroboros3D | 7.93% | 10.00% |
> | Era3D | 22.33% | 17.20% |
> | **Ours (SD2.1)** | **25.07%** | **36.80%** |
>
> **3. Use of Existing Components.**
>
> We acknowledge that components like multi-view self-attention and reference attention have been used in prior work. However, our core contribution lies in our **innovative adapter-based framework** for multi-view image generation, not just in individual components. Specifically:
> - **First Adapter-Based Solution:** We introduce the first adapter framework for this task, achieving efficiency, versatility, and adaptability.
> - **Innovative Adapter Framework Design:** Our design includes a novel parallel attention architecture and a unified condition embedding and encoder, which enables the versatility of MV-Adapter.
> - **Extensions to Arbitrary View Generation and Inspirations for Future Work:** MV-Adapter can extend to **arbitrary view generation**, and provides insights for modeling geometric, physical or temporal knowledge based on image priors with our framework for decoupled learning.
>
> We kindly refer you to our global response for further elaboration.
>
> **4. Details on Experimental Setup for GSO Dataset Comparison.**
>
> For the image-to-multiview comparison on the GSO dataset, we selected 100 assets covering multiple object categories. For each asset, we rendered an input image from a frontal view within azimuth angles of -45° to 45° and elevation angles of -10° to 30°. The evaluation views did not include extreme top or bottom angles. Additional details are provided in **"Comparison with baselines" of  Appendix A.2**.
>
> We hope these clarifications address your concerns. Thank you again for your valuable feedback.

---

> ### Comment · Reviewer_v3NF · 2024-11-21
> **Thank you for your rebuttal**
>
> It's a really nice rebuttal with tons of results and new visualizations. Very impressive job!
>
> Overall, I am happy with every point in the rebuttal, and many of my concerns are well-addressed. Though I remain concerned about the incremental nature and novelty, the performance aspect is now mostly in very good shape with sufficient validation.
>
> As stated in my review, my initial assessment was more like a 4 (weak reject); now I am inclined to change it to a 5 (borderline reject). I cannot assign a higher score due to the high standards of ICLR.

---

> > ### Author Response · Authors · 2024-11-22
> > **Regarding Novelty and Meeting ICLR Standards (1)**
> >
> > Thank you for your positive remarks and acknowledgment of our "impressive" additional results. We appreciate that our additional results and clarifications have helped address many of your initial concerns.
> >
> > **1. Regarding Novelty and Incremental Nature**
> >
> > We observe that your initial review mentioned component novelty only as a "minor" concern, specifically noting that "The condition guider is inspired by T2I-Adapter, and the decoupled attention borrows from Era3D." However, this has now become a major factor in your assessment. We would like to emphasize several substantial technical innovations that distinguish our work:
> >
> > 1. **First Systematic Adapter Framework for Multi-View Generation.**
> > We introduce the first comprehensive adapter-based framework specifically designed for multi-view image generation. While adapters have been used in other contexts, applying them effectively to multi-view generation is non-trivial due to the complexities involved in handling multiple views and ensuring consistency. Our framework leverages the strengths of adapters—such as efficiency, versatility, and adaptability—to address the challenges inherent in multi-view image generation. This approach allows us to process high-resolution images (up to 768×768) and to scale with larger base models like SDXL, which was previously infeasible with existing methods.
> > 2. **Novel Technical Architecture.** In response to your initial comment that _"The condition guider is inspired by T2I-Adapter, and the decoupled attention borrows from Era3D"_, we would like to emphasize that our main contribution lies in the **innovative adapter framework** we propose for multi-view image generation, rather than in specific components.  Our adapter design incorporates two key technical innovations:
> >     - **Attention Architecture:** Our design adheres to the principle of **preserving the original network structure and feature space of the base T2I model**. To achieve this, we duplicate the self-attention layers (including both architecture and weights) to create new multi-view attention and image cross-attention layers while initializing the output projections to zero, allowing the new layers to learn geometric knowledge without interfering with the original model. Furthermore, we enhance the effectiveness of our attention layers through a **parallel organization structure**. Unlike existing methods that modify the base network and disrupt learned priors (e.g., MVDream and Zero123++), our approach maintains the integrity of the base model, enabling efficient training and better performance. Our ablation results in **Table 5 and Figure 17** directly demonstrate the superiority of this design. Detailed discussions can be found in our global response.
> >     - **Unified Condition Embedding and Encoder:** We innovatively unify camera parameters and geometric information into spatial map representations, introducing a spatial-aligned condition encoder that handles both types of conditions using the same architecture. This unification enhances the flexibility and applicability of our method, allowing it to support multiple functionalities seamlessly—a capability not achieved by previous works.
> >
> > Notably, our adapter architecture is not limited to specific components like row-wise attention; as shown in our newly added arbitrary view image generation experiments, it can be extended to incorporate various attention mechanisms for more applications.
> >
> > Overall, our contributions are significant both in terms of methodology and performance improvements.

---

> > ### Author Response · Authors · 2024-11-22
> > **Regarding Novelty and Meeting ICLR Standards (2)**
> >
> > **2. Meeting ICLR Standards**
> >
> > We believe our work aligns with ICLR's core emphasis on bringing _"new, relevant, impactful knowledge"_ to the community through:
> > - **Novel Technical Contribution:**
> >     1. **First Systematic Adapter Framework for Multi-View Generation:** We introduce the first comprehensive adapter-based framework specifically designed for multi-view image generation, effectively addressing the complexities of handling multiple views and ensuring consistency.
> >     2. **Innovative Design:** Our attention architecture and unified condition embedding enable efficient integration of geometric knowledge without altering pre-trained models.
> >     3. **Thorough Ablation Studies:** We provide extensive ablations validating our design choices, demonstrating their impact on performance.
> > - **Well-motivated Problem & State-of-the-Art Performance:**
> >     1. **Addresses critical limitations:** Our work tackles significant limitations in existing methods, such as adaptability, scalability and resolution constraints, enabling high-resolution (768×768) multi-view generation on large models like SDXL.
> >     2. **SOTA Results:** We achieve leading performance in multi-view generation, with improved consistency and quality.
> > - **Impact and Utility for the Community:**
> >     1. **Adaptability and Versatility:** Our MV-Adapter seamlessly integrates with various powerful or customized T2I model variants, enabling consistent multi-view generation and supporting multiple conditional inputs, which facilitates downstream applications like 3D generation and 3D texture synthesis.
> >     2. **Inspiring Future Research:** Our approach of learning new knowledge based on existing priors can inspire the modeling of other geometric (e.g., lighting effects), physical, and temporal knowledge, potentially benefiting a wide range of applications.
> >
> > Given that we have comprehensively addressed your initial major concerns (arbitrary view capability, stronger quantitative validation) and demonstrated substantial technical novelty, we respectfully request reconsideration of your assessment of our paper.
> >
> > Thank you once again for your thoughtful feedback. If you have any further concerns or questions, we would be happy to address them.

---

> > > ### Comment · Reviewer_v3NF · 2024-11-25
> > >
> > > Thanks to the authors for the second-round discussion and further clarifications.
> > >
> > > I don't have major concerns regarding this work; it's a borderline one. Accepting it is OK, and rejecting it won't be very harmful either. Given the very solid experiments presented and valuable insights for future research, it might be better to lean a bit more towards acceptance, so I would like to raise my score to a 6.

---

### Official Review · Reviewer_Fnks · 2024-11-02

**Soundness:** 3
**Presentation:** 4
**Contribution:** 3
**Rating:** 6
**Confidence:** 4

**Summary:**

The paper proposes a novel module called MV-Adapter, designed as a plug-and-play solution for enhancing pre-trained text-to-image (T2I) models in multi-view image generation.

MV-Adapter addresses key limitations of existing methods, such as high computational costs, incompatibility with model derivatives, and limited versatility. By integrating with T2I models without invasive modifications or full-parameter training, MV-Adapter enables efficient high-resolution synthesis while maintaining compatibility with various extensions and derivatives like personalized models and plugins.

It supports diverse conditioning signals, facilitating applications in text and image-based 3D generation and texturing. Experiments show that MV-Adapter sets a new standard in generating multi-view consistent images, combining efficiency, adaptability, versatility, and high performance.

**Strengths:**

- MV-Adapter streamlines the training process by eliminating full fine-tuning, enabling efficient high-resolution image generation with reduced computational costs.

- MV-Adapter seamlessly integrates with various derivatives and extensions of the base T2I model, such as personalized models and plugins like ControlNets, enhancing its versatility across different applications.

- MV-Adapter supports diverse conditioning inputs (e.g., text, images, geometry), and MV-Adapter broadens the scope of multi-view generation and ensures consistent, high-quality output across various tasks.

**Weaknesses:**

I would like to positively acknowledge that the results are impressive.
- However, the weakness I found in this paper is that the contributions seem somewhat limited. The main contribution appears to be the application of self-attention, and if there are other contributions, could you revise the paper to emphasize those points? It would be helpful to clearly explain what differentiates this work from existing self-attention methods in multi-view generation, and what new methods, beyond self-attention, have been introduced in this paper.

- Additionally, I believe that including an evaluation comparing parallel and serial architectures would make this a stronger paper. Conducting an ablation study to compare the proposed parallel architecture with a serial architecture and evaluating generation quality, multi-view consistency, and computational efficiency would be beneficial.

- Overall, since the consistency of the results is good, enhancing the novelty aspect would make this a better paper. What is the novelty of this paper compared to existing multi-view generation methodologies? Furthermore, mentioning the potential applicability of this paper and highlighting potential new applications would also be helpful.

**Questions:**

It has been stated in the Weaknesses section. I hope the authors will address these points in the rebuttal session.

---

> ### Author Response · Authors · 2024-11-21
>
> Thank you for acknowledging our impressive results.
>
> **1. Clarify and Emphasize Novel Contributions.**
>
> Our key contributions extend beyond merely applying self-attention and are outlined below: (1) We introduce **the first adapter framework** specifically designed for **multi-view image generation**. This framework offers efficiency, adaptability, and versatility by integrating geometric knowledge without altering pre-trained T2I models, demonstrating high extensibility and performance. (2) Our **innovative adapter framework** features an effective parallel attention architecture and a unified camera and geometric condition encoder. These innovations enable effective decoupling and integration of diverse conditions, enhancing the overall performance and flexibility of the model. (3) MV-Adapter can extend to **arbitrary view generation**, and provides **insights** for modeling geometric, physical or temporal knowledge based on image priors with our framework for decoupled learning.
>
> We are revising the paper to emphasize these contributions and refer you to **our global response** for detailed explanations.
>
> **2. Evaluation Comparing Parallel and Serial Architectures.**
>
> We provided qualitative comparisons in the original version, and have added quantitative comparisons and additional qualitative results in **Table 5 and Figure 17** of the revised paper, demonstrating the advantages of our parallel design.
> > Table 5
> | Attention Architecture (Based on SDXL) | PSNR $\uparrow$ | SSIM $\uparrow$ | LPIPS $\downarrow$ |
> | - | :-: | :-: | :-: |
> | Serial | 20.687 | 0.8681 | 0.1149 |
> | **Parallel (Ours)** | **22.131** | **0.8816** | **0.1002** |
>
> **3. Enhance the Novelty Aspect Compared to Existing Methods.**
>
> Our method offers distinct contributions: 1) an efficient adapter framework design that decouples geometric knowledge modeling from the base T2I models in a non-invasive manner, which preserves the image priors and enables efficient training on larger models; 2) improved usability and customizability, which means higher generation quality, applicability to larger base models like SDXL, strong versatility and adaptability to various base model variants.
>
> **4. Highlight Potential Applications.**
>
> MV-Adapter's adaptability enables new applications in image creation and personalization, such as integrating with personalized T2I models for customization. For instance, creators can integrate MV-Adapter with their personalized T2I models—customized for specific identities or artistic styles—to generate multi-view images that capture consistent perspectives of their unique concepts. Furthermore, Its successful practice of decoupling image prior and geometric knowledge also provides insights for tasks requiring modeling of geometry (like lighting, zoom in/out) or other novel aspects (like physics or temporal knowledge) based on T2I models (see **Appendix A3.5**).
>
> We hope that our responses address your concerns and clarify the contributions and novelty of our work. Thank you again for your valuable feedback.

---

> ### Author Response · Authors · 2024-11-25
>
> Dear Reviewer:
>
> Once again, we sincerely appreciate the time and effort you have dedicated to reviewing our paper!
>
> As the discussion period concludes in two days, we would be grateful if you could review our rebuttal at your convenience, should your schedule allow. If there are any further points requiring clarification or improvement, please be assured that we are fully committed to addressing them promptly. Thank you once again for your invaluable feedbacks to our research!
>
> Warm regards, The Authors

---

> ### Comment · Reviewer_Fnks · 2024-11-25
>
> Dear Authors,
>
> Thank you for your response. It seems that most of the points I mentioned have been addressed, and I sincerely appreciate the authors' efforts. I will maintain my original rating ('6: marginally above the acceptance threshold').
>
> Best regards,
> Reviewer Fnks

---

### Author Response · Authors · 2024-11-21
**Global Response: Highlighting Our Key Contributions (2)**

**2. Technical Innovations in Adapter Design**

Our adapter framework incorporates the following two key technical contributions:

**Attention Architecture.** Building on the importance of adapters for the multi-view generation task, our design adheres to the principle of preserving the original network structure and feature space of the base T2I model. To achieve this, we duplicate the self-attention layers (including both architecture and weights) to create new multi-view attention and image cross-attention layers. By initializing the output projections to zero, we ensure that the base model's priors are maintained while allowing the new layers to learn geometric knowledge without interfering with the original model. In contrast, existing methods like MVDream expand the keys and values of the base model's original self-attention layers to include multi-view features, and Zero123++ similarly extends the key and value scopes to incorporate features from the reference image. These existing approaches modify the original network and can disrupt the learned priors, making them incompatible with the adapter principle.

Furthermore, we enhance the effectiveness of our attention layers through a parallel organization structure. The most straightforward way to insert new network layers is to append them after the original layers, connecting them in a serial manner. However, this sequential arrangement may not effectively utilize the image priors modeled by the pre-trained self-attention layers, as it requires the new layers to learn from scratch. Even if we initialize the new layers with the pre-trained self-attention parameters, the features input to these new layers are in a different domain, causing the initialization priors to be ineffective. To fully exploit the effective priors of the spatial self-attention layers, we design a parallel architecture. This design ensures that the new attention layers can inherit powerful prior to learn geometric knowledge efficiently. Detailed discussions of this design are presented in the **"Attention Architecture" section of our paper (Section 4.2)**. Our ablation results in **Table 5** directly demonstrate the superiority of this design.

**Unified Condition Embedding and Encoder.** To enable the adapter's versatility—supporting both camera conditions and geometry conditions for critical 3D applications like 3D generation and 3D texture generation—we innovatively unify camera parameters and geometric information into spatial map representations. We introduce a spatial-aligned condition encoder that encodes both types of maps using the same architecture. Previous methods have only encoded either camera information or geometric information but have not unified both within a single framework.

By adopting this unified representation and encoder, our adapter gains enhanced flexibility and applicability, seamlessly handling various types of guidance within a unified framework. This design allows the model to support multiple functionalities without additional complexity, which previous works have not achieved.

**3. Extensions to Arbitrary View Generation and Inspirations for Future Work**

**Extensibility to Arbitrary View Generation.** Our method is not limited to generating a fixed set of six views; it can be extended to arbitrary view generation, enabling broader applications. We achieve this by first generating multiple anchor views and then using them as conditions to guide the generation of additional views at arbitrary angles. The MV-Adapter framework remains largely unchanged—we adjust the number of input images to one or four (concatenating multiple images into a long image if needed) and replace the row-wise attention with full self-attention in the multi-view attention module. Detailed schemes and experimental results are provided in **Appendix A3.6** and on **our anonymous project page** https://mv-adapter.github.io/.

**Learning New Knowledge Like MV-Adapter.** By decoupling the learning of geometric knowledge from the image prior, our framework efficiently integrates new knowledge without compromising the base model's rich visual capabilities. This principle not only enhances learning from limited data but also inspires other tasks that build upon existing image priors to learn new types of knowledge. For example, beyond multi-view consistency, our approach can be applied to learning zoom in/out effects, consistent lighting, and other viewpoint-dependent properties. It also provides insights for modeling physical or temporal knowledge based on image priors.

---

### Author Response · Authors · 2024-11-21
**Global Response: Highlighting Our Key Contributions (1)**

We thank all reviewers for their constructive feedback and for recognizing the strengths of our work. Specifically, we appreciate Reviewer Fnks' comment that "the results are impressive," as well as Reviewer v3NF and Reviewer C83o's remarks that the paper is "well-written" and that "experiments are solid." These positive remarks encourage us to further clarify and strengthen the key contributions of our work.

We kindly suggest that reviewers visit **our anonymous project page** https://mv-adapter.github.io/ for a deeper understanding of our work. We note that this project page was already provided in our initial submission's supplementary materials for reference.

Additionally, we have uploaded the revised paper and included a highlighted version of the paper (with updated content highlighted) in the supplementary materials to facilitate the reviewers' review.

The primary concern raised is the lack of technical innovation in our adapter design. We would like to address this by highlighting the following key points:

**1. Novel Contribution: First Adapter-Based Solution for Multi-View Image Generation**

The reviewers questioned that the **components** of our method are not very new — this is true — multi-view self-attention, reference attention and spatial-aligned control are not new and have been widely used in prior literatures across different tasks. However, what we’d really like to address as our major contribution, is the design of unifying these techniques as an effective **“adapter”** framework working on top of pre-trained T2I models for the **multi-view image generation task**. Here we’ll show why “adapters” are so important especially for this task we want to solve.

**Why adapters?** Adapters are plug-and-play modules designed to work seamlessly with pre-trained generation models, facilitating downstream tasks by introducing external control signals or constraints. Crucially, they achieve this without invasive changes to the pre-trained models, thereby preserving the prior knowledge encoded in the base models. To adapt a pre-trained model for new tasks, we can always make changes to the network structure and do full fine-tuning. Then why would we need adapters? There are several good properties of adapters which we would favor against full fine-tuning:
- **Adapters are easy to train (efficiency).** Adapters only require updating a small number of parameters, making them both faster to train and memory-efficient. This property has become increasingly critical as state-of-the-art T2I models grow in scale (e.g., ~800M parameters in SD1.5 to ~12B in Flux.1), making full fine-tuning infeasible for smaller research labs.
- **Adapters keep versatility.** Downstream tasks often have limited training data compared to the billions of (text, image) pairs used to train base T2I models. Full fine-tuning on such limited datasets risks overfitting and mode collapse, impairing the generalization capability of the models. Adapters mitigate this risk by constraining the optimization space through fewer trainable parameters.
- **Adapters are easy to use (adaptability).** Adapters are plug-and-play modules and can be directly applied to different variants of base models, including customized fine-tuned versions and LoRAs. They can also interoperate with other adapters, enabling the construction of flexible workflows that accommodate diverse control signals. This is the most important reason why adapters are actively being developed and supported by the community.

These properties make adapters particularly suitable and useful for multi-view image generation.

**Relevance to Multi-view Image Generation.** Generating view-consistent images involves interaction across multiple views, requiring at least N images to be processed simultaneously during training. This is computationally challenging when working with large base models and high-resolution images. Moreover, the scarcity of 3D training data exacerbates optimization difficulties when performing full model fine-tuning. Previous multi-view generation methods are limited to resolutions of 256x256 (MVDream) or 384x384 (Zero123+++) and struggle with resolutions of 512x512 (Instant3D, Era3D). In contrast, modern T2I models can achieve resolutions of 1024x1024 or higher. These limitations primarily stem from the challenges associated with full training.

By leveraging our adapter design, we successfully achieve 768x768 resolution multi-view generation on the SDXL model. As shown in our paper, our trained MVAdapter demonstrates versatility, enabling seamless application to base model variations for stylized or customized multi-view generation, and compatibility with other adapters such as ControlNets.

**The above effects are achieved from our effective adapter framework, and the following innovative design in our framework.**

---

### Public Comment · ~Yoonwoo_Jeong1 · 2024-12-02
**Request for acknowledgment and comparison with NVS-Adapter(ECCV2024) which includes significant overlaps with MV-Adapter.**

I appreciate your impressive and remarkable results in text-to-3D generation tasks.
I am the first author of "NVS-Adapter: Plug-and-Play Novel View Synthesis from a Single Image," which was presented at ECCV 2024 ([link](https://eccv.ecva.net/virtual/2024/poster/1425)). After reading your paper, we noticed that it includes significant overlaps with our core contributions but does not discuss or mention our work at all.

Our initial version has been available on arXiv since 2023 ([link](https://arxiv.org/abs/2312.07315)), and our code is also publicly accessible as an open-source project ([link](https://github.com/kakaobrain/nvs-adapter)). Thus, I believe our work cannot be considered a concurrent effort to your study and SHOULD be mentioned and thoroughly compared. Moreover, considering that all reviewers raised concerns about your paper’s technical novelty, I strongly believe that comparing your study with existing approaches will be a critical factor in evaluating its contributions.

For your reference, the core contributions of NVS-Adapter can be summarized as follows:

1. NVS-Adapter is the first approach to adopt the concept of Adapters for synthesizing multiple novel views in 3D generation.
2. Novel multi-views are generated simultaneously using attention-based architectures.
3. The plug-and-play approach enables a T2I model to be compatible with existing modules, such as ControlNet and LoRA, for various 3D generation scenarios, without requiring additional fine-tuning.

Thank you for considering this feedback.

---

> ### Author Response · Authors · 2024-12-03
>
> Dear Yoonwoo Jeong,
>
> Thank you very much for your thoughtful feedback and for bringing the overlaps with NVS-Adapter to our attention. We sincerely apologize for the oversight and appreciate your contribution to this field. We fully agree that acknowledging and comparing our work with NVS-Adapter is important for providing a comprehensive discussion of the related approaches.
>
> After carefully reading and comparing the two versions of your paper—the initial arXiv version (2312.07315v1) and the latest ECCV 2024 version (posted on 10 Aug 2024)—we have identified several key points that we believe highlight the differences between NVS-Adapter and MV-Adapter.
>
> **Different Versions of NVS-Adapter**
>
> 1. **Version 1 (arXiv, 2023):** The primary motivation and contribution of NVS-Adapter in this version focus on "synthesizing novel multi-views of visual objects while **fully exploiting the generalization capacity of T2I models**." This version does not consider the compatibility with LoRA, which was mentioned in your comment. As such, we do not see "significant overlaps" between this version of NVS-Adapter and our work, as the emphasis in NVS-Adapter is more on the generalization aspect of T2I models, which is a common issue that many existing works have focused on.
> 2. **Version 2 (August 2024):** The latest version of NVS-Adapter, which is published in August 2024 and **should be considered as a concurrent work** according to ICLR's policy, aligns more closely with the comment you provided, focusing on "compatibility with existing modules such as ControlNet and LoRA". However, the compatibility with T2I extensions is only part of the adaptability of MV-Adapter.
>
> **Differences and Our Distinct Contributions**
>
> **1. Task Scope:**
> * Multi-view generation refers to the process of generating multiple images of the same object or scene from different viewpoints. It is a more generalized task that involves synthesizing consistent views based on a variety of input conditions (text, images, camera parameters, or geometry). In this broader context, **MV-Adapter is designed to tackle multi-view consistent image generation with flexible conditions, which is not limited to a more specific task of synthesizing novel views from a single image (as in NVS-Adapter)**. This makes MV-Adapter a **first adapter-based solution** designed to meet **a wide range of multi-view generation needs**.
> * We believe this distinction in task scope sets our work apart and justifies calling MV-Adapter **the first systematic adapter solution for multi-view image generation**.
>
> **2. Motivation and Focus:**
> * The primary focus of NVS-Adapter (version 1) is on synthesizing novel views while **fully exploiting the generalization capacity of T2I models**, which involves freezing the base model. The version 2 focuses on the compatibility with T2I extensions, which is only part of the adaptability of MV-Adapter. Our work emphasizes **efficiency, adaptability, and versatility** through an adapter-based framework. We prioritize not just generating novel views but also ensuring that our approach can seamlessly handle various conditions (text, image, and geometry) and scale efficiently.
> * While NVS-Adapter (version 2) considers a certain degree of adaptability, MV-Adapter's **complete adaptability, scalability, and versatility in supporting diverse input conditions** remain key differentiators.
>
> **3. Methodology:**
> * NVS-Adapter freezes the base model and models view consistency using an additional attention-based network, which can only be trained from scratch and leads to limited effectiveness. In contrast, MV-Adapter **duplicates and augments** the self-attention layers of the base T2I model, employing a **parallel architecture** for multi-view attention and image cross-attention. This allows our model to leverage the pre-existing priors to learn view consistency and reference knowledge without disruptive retraining, thus boosting efficiency and adaptability. Additionally, we introduces a **unified encoding mechanism** for camera and geometric conditions, enabling broader applicability and making the approach more flexible and versatile.
> * The above comparison **further confirms that the effectiveness of our proposed technical innovations** in our architecture, which allows MV-Adapter to seamlessly integrate with various T2I models with their extensions and generate high-quality and consistent multi-view images.
>
> **Acknowledgment**
>
> We will certainly address the relationship between NVS-Adapter and MV-Adapter in the revised version of our paper, providing a more thorough comparison. We will highlight the distinctions in task scope, motivation, methodology, overall design, and generated results.
> Once again, we apologize for not addressing this overlap earlier, and we thank you for your constructive feedback. We are grateful for the opportunity to improve the clarity of our work and to acknowledge the contributions of NVS-Adapter.
>
> Warm regards,
>
> The Authors

---

> ### Public Comment · ~Doyup_Lee1 · 2024-12-03
>
> Dear Authors.
>
> Hello, I am Doyup Lee who is one of the corresponding authors of NVS-Adapter (ECCV'24). First, I sincerely appreciate the nice and interesting idea for versatile multi-view generation based on Adapter, and the authors' responses on our public comment. However, I want to discuss some of points more to clarify the contribution of this paper based on my understanding.
>
> > NVS-Adapter focuses on generating novel views from a single input view (NVS). In contrast, MV-Adapter addresses a much broader task, tackling multi-view consistent image generation with flexible conditions such as text, image, and geometric inputs.
>
> Our NVS-Adapter also includes the experiments to generate multi-views and 3D assets also from user-provided text prompt.
>
> > This makes MV-Adapter a first adapter-based solution designed to meet a wide range of multi-view generation needs.
>
> The authors should not claim that it is the "first" adapter-based solution as the originality, following ICLR's Code of Ethics, "Researchers should therefore credit the creators of ideas, inventions, work, and artefacts, and respect copyrights, patents, trade secrets, license agreements, and other methods of protecting authors' works." If the authors want to claim the contributions, comparing with our first version, this paper's main contributions should be elaborating and extending the adapter-based multi-view generation method, which is previously proposed.
>
> Meanwhile, I strongly believe that our second version, published in ECCV'24, is a "natural" extension from the first version, not a proposal of new value. Note that the core value of our NVS-Adapter is not the number of experiments, but lies in"composability and generalization ability". In addition, the original paper of T2I Adapter clearly mentions "Further, the proposed T2I-Adapters have attractive properties of practical value, such as composability and generalization ability." in its abstract. Although our first version lacks enough experiments to show how to fully enjoy the "composability and generalization," our second version is a natural extension to add more experiments, keeping the original value "how adapter-based multi-view generation can support various scenarios without fine-tuning of existing T2I models."
>
>
> > 3. Methodology
>
> I agree that the paper has a difference in the model architecture, and I like the core idea of parallel attentions. Meanwhile, note that NVS-Adapter's `View-Consistency Cross-Attention` include self-attention of each target view, because learnable tokens aggregate all information of target views. I also agree that NVS-Adapter do not have ` camera and geometric conditions` but only have `camera conditions`.
>
> > should be considered as a concurrent work according to ICLR's policy,
>
> Note that the ICLR policy also recommend "Authors are encouraged to cite and discuss all relevant papers". In addition, I'm surprised that the authors failed to find our paper, **although the "naming" and "contributions" have severe overlaps** and our first version's paper and code **has been available from the last year**. For you reference, our "published" paper was available 52 days before the ICLR submission.

---

> ### Author Response · Authors · 2024-12-03
>
> Dear Dr. Lee,
>
> Thank you for your feedback. We will further clarify the contributions of our work as follows.
>
> **1. Claim of MV-Adapter as the First Systematic Adapter-Based Solution for Multi-View Image Generation**
>
> We acknowledge your work on the NVS-Adapter and its contributions to **novel view synthesis from a single input view**. Our claim that MV-Adapter is the first systematic adapter-based solution for multi-view image generation is based on the broader scope of the definition of multi-view generation task. While NVS-Adapter focuses primarily on generating novel views from a single input view, **MV-Adapter addresses a more generalized multi-view generation task** that incorporates flexible conditions such as text, image, and geometric inputs, positioning it as a pioneering solution in this more comprehensive context.
>
> Although, as you mentioned, your work also includes experiments to generate multi-views from text prompts, the approach involves first generating images from text and then generating novel views from these images, where NVS-Adapter only works for the latter stage (i.e. novel view synthesis from single view). Therefore, NVS-Adapter serves as an adapter for a specific task that is **novel view synthesis from a single view**.
>
> Moreover, we emphasize that **the text-conditioned adapter significantly enhances flexibility and adaptability**. It allows MV-Adapter to integrate with different base model variants, such as personalized text-to-image models, achieving more applicable multi-view personalization or customization. In contrast, NVS-Adapter shows inflexible due to the limitation to image conditions, which inherently impose stronger appearance constraints. This limitation reduces the applicatibility of the model, as the generated results rely heavily on the predefined visual characteristics of the input images. For example, when generating multi-view images from a single image, it struggles to produce corresponding styles or customized concepts even using personalized LoRA. In contrast, our MV-Adapter is capable of generating multi-view consistent and customized images, thereby providing greater benefits to the community.
>
> **2. On the Evolution of NVS-Adapter from Version 1 to Version 2**
>
> Regarding your assertion that version 2 of NVS-Adapter is a natural extension of version 1 and these two versions introduce the same advantages on adaptability, we respectfully disagree. Version 1 of NVS-Adapter lacks comprehensive adaptability experiments, which is difficult to reflect the characteristics of the "adapter". Therefore, version 2 is a significant improvement over version 1. Furthermore, even the adaptability experiments in version 2, such as compatibility with T2I extensions like ControlNet and LoRA, constitute **only a portion of the adaptability aspects** addressed in our MV-Adapter. Importantly, our work introduces a text-to-multiview adapter mechanism, a task setting that has not been addressed in your NVS-Adapter. This represents a significant advancement beyond what is covered in both versions of NVS-Adapter, thereby highlighting the novel contributions of our MV-Adapter.
>
> **3. Concurrent Work and Citation**
>
> We acknowledge the importance of recognizing concurrent work and adhering to ICLR’s policy on citing and discussing relevant papers. We regret any oversight in our initial submission and will ensure that our revised version includes a comprehensive discussion and citation of your work on NVS-Adapter. This will provide a clearer context for our contributions and highlight the distinct advancements our MV-Adapter offers.
>
> Once again, thank you for your valuable feedback. We are committed to refining our work to accurately reflect its contributions and to honor the advancements made by your research.
>
> Best regards,
>
> The Authors

---

> > ### Public Comment · ~Doyup_Lee1 · 2024-12-03
> >
> > I sincerely appreciate the quick responses. For clarification, I humbly left my opinions about the authors' response to clarify the controversial points and help refining the authors' work accurately reflect them.
> >
> > >  Therefore, NVS-Adapter serves as an adapter for a specific task that is novel view synthesis from a single view.
> >
> > I agree the contribution of MV-Adapter as targeting a generalized multi-view task. Our NVS-adapter "module" is designed to conduct novel "multi-view" synthesis from a single image. However, thanks to the nature of adapter-based approach, the "framework" of NVS-Adapter can successfully conduct text-to-multi-view generation as shown in our experiments. That is the reason why we incorporate the adapter-based approach for a minimal module addition to fully leverage T2I models' own capability and adapt them to new tasks, **as the same claim** with this paper. Thus, we don't need to interpret the scope of tasks too narrowly.
> >
> > > 2. On the Evolution of NVS-Adapter from Version 1 to Version 2. Regarding your assertion that version 2 of NVS-Adapter is a natural extension of version 1 and these two versions introduce the same advantages on adaptability, we respectfully disagree. Version 1 of NVS-Adapter lacks comprehensive adaptability experiments, which is difficult to reflect the characteristics of the "adapter".
> >
> > I agree that the viewpoint can be different by readers, but I still think that it is a natural extension to enhance our first version in terms of predictability, and I feel the authors are trying to over-differentiate NVS-Adapter with the proposed MV-Adapter. It would be helpful to refine the revised version based on the facts below:
> > - First, the plug-and-play property itself reflects the characteristics of the "adapter". The first version already shows the adaptability of the plug-and-play NVS-adapter to T2I model for novel "multi-view" synthesis, enabling image-to-3D or text-to-3D generation enough.
> > - Second, while version 2 focuses on adding more experiments about compatibility with other existing modules, note that using multiple adapters at once (to fuse various tasks) or using an adapter with LoRAs has been **a well-known idea** in the community. For example, `StableDiffusionAdapterPipeline` already supports `List[T2IAdapter]` as its parameter: https://huggingface.co/docs/diffusers/main/en/api/pipelines/stable_diffusion/adapter .
> >
> >
> > Again, I appreciate your prompt responses and detailed explanation.
> > I hope my comments help improving the revised version of MV-Adapter.
> >
> > Best.

---

> > > ### Author Response · Authors · 2024-12-03
> > >
> > > Thank you once again for your detailed feedback on our paper submission.
> > >
> > > > Our NVS-adapter "module" is designed to conduct novel "multi-view" synthesis from a single image. However, thanks to the nature of adapter-based approach, the "framework" of NVS-Adapter can successfully conduct text-to-multi-view generation as shown in our experiments.
> > >
> > > We believe there is a slight misinterpretation. Although it is possible to first use a text-to-image generation model to convert text to an image and then **use NVS-Adapter to generate novel views from that image**, it cannot demonstrate that NVS-Adapter can be inherently a text-to-multi-view generation model.
> > >
> > > > I agree with the contribution of MV-Adapter as targeting a generalized multi-view task.
> > >
> > > Based on our task definition and your agreement, we firmly believe that **MV-Adapter is the first systematic adapter-based solution for multi-view image generation**, which incorporates flexible conditions such as text, image or geometric inputs. This broader scope positions MV-Adapter as a pioneering solution in this more comprehensive context.
> > >
> > > > It would be helpful to refine the revised version based on the facts below.
> > >
> > > Of course, we will ensure that the revised version of our paper fully acknowledges the contributions of NVS-Adapter in the task of novel view synthesis from single images, and its compatibility with other T2I modules like ControlNet and LoRA.
> > >
> > > Thank you again for your valuable insights, which have been instrumental in refining our work.

---

### Author Response · Authors · 2024-12-04
**Conclusion**

First and foremost, we extend our sincere gratitude to the reviewers and the public commenters for their detailed, insightful feedback and engaging discussions. Below, we summarize the key developments and reinforce the core contributions of our paper during the rebuttal phase.

**1. Enhancements Addressed During the Rebuttal Phase**

- **Validation of Technical Innovations:** We have conducted additional quantitative analyses and provided more extensive qualitative visualizations to substantiate the effectiveness of our proposed parallel architecture.
- **Comprehensive Evaluation of MV-Adapter:** We incorporated a user study within the image-conditioned multi-view generation setting and evaluated the geometry quality of 3D reconstruction. These evaluations confirm that MV-Adapter achieves state-of-the-art consistency and quality in multi-view generation.
- **Extension to Arbitrary Viewpoint Generation:** We have successfully extended MV-Adapter to support arbitrary viewpoint generation, accompanied by a substantial number of visualizations. This extension verifies the scalability and broad applicability of MV-Adapter across diverse scenarios.
- **Exploration of New Applications and Future Works:** MV-Adapter's compatibility with various personalized models and spatial control plugins allows for multi-view image customization and controllable generation. Additionally, by decoupling the learning of geometric knowledge from the image prior, MV-Adapter presents a more effective pathway for modeling new knowledge, such as physical or temporal aspects.
- **Supplementary Technical Details:** We have enriched the technical exposition of our paper by elaborating on the image cross-attention mechanisms, detailing the calculation of CFG with multiple conditions, and clarifying the experimental settings for comparisons with baseline methods.

**2. Summary of Responses to Public Comments**

Based on discussions and consensus with the authors of NVS-Adapter, we further confirmed one of our core contributions: presenting the first systematic adapter-based solution for multi-view image generation. We also acknowledge the oversight in not citing and discussing NVS-Adapter adequately in our original submission. We commit to addressing this in the revised version by including a comprehensive discussion of NVS-Adapter and its relation to our work.

**3. Reiteration of Novelty and Core Contributions**

- We introduce **the first systematic adapter-based solution for multi-view image generation**, significantly enhancing its efficiency, adaptability, and versatility.
- **Technical Innovations in Adapter Design:** Our innovative attention architecture, which includes the duplication of self-attention layers and a parallel architecture, along with the unified condition encoder that supports encoding camera and geometry conditions, represents a substantial advancement. Ablation studies have validated the effectiveness of these innovations. We appreciate the recognition of our technical contributions by the public commenters.
- **Extensions to Arbitrary View Generation and Future Inspirations:** The ability of MV-Adapter to effortlessly extend to arbitrary viewpoints paves the way for numerous downstream applications. Furthermore, our approach to decoupling geometric knowledge learning from image priors offers a novel and efficient methodology for modeling new types of knowledge.

In summary, the rebuttal phase has allowed us to strengthen our paper by providing additional validations, expanding our evaluations, and clarifying our contributions. We are confident that these enhancements, along with our novel contributions, substantiate the significance and impact of our work in the field of multi-view image generation.

---

### Meta-Review · Area_Chair_JGmr · 2024-12-19

**Metareview:**

This submission proposes an adapter-based method for consistent multi-view image generation. The paper received reviews from five reviewers resulting in five borderline scores. Multiple reviewers recognised the comprehensiveness of the experimental work and quality of writing and presentation. The core of the main concerns related to a lack of technical innovation and comments on incremental novelty.

The author rebuttal could partially address concerns by presenting additional evaluation results and explanations enabling two reviewers to initially raise their scores after extended discussion. Further to this, public comments highlighted that recent highly related work was unacknowledged. The initial failure to acknowledge the relevant NVS-Adapter work, coupled with a somewhat unconvincing distinction justification did not convince some reviewers, causing multiple reviewers to lean towards rejection in the private discussion phase, based on core unresolved concerns regarding incremental contributions and lack of technical innovation in adapter design.

Authors noted their intention to address the relationship between NVS-Adapter and MV-Adapter. Paper modifications are proposed towards providing a thorough comparison and highlighting distinctions, however the proposed changes appear somewhat substantial ('task scope, motivation, methodology, overall design, and generated results'). There was a consensus that such additions are necessary and yet are currently not present in the manuscript version under review.

**Additional Comments On Reviewer Discussion:**

This submission was discussed at length by the AC, SAC and PCs.

A consensus was reached that without appropriate treatement of the noted related work, contributions can appear incremental in nature and that any updated manuscript, rectifying this point, likely benefits from a further round of review. The consensus is therefore to recommend rejection at this stage.

---

### Decision · Program_Chairs · 2025-01-22

Reject